# Deep Networks Learn Deep Hierarchical Models

**Amit Daniely** [1]

## Abstract

We consider supervised learning with $n$ labels and show that layerwise SGD on residual networks can efficiently learn a class of hierarchical models. This model class assumes the existence of an (unknown) label hierarchy $L_1 \subseteq L_2 \subseteq \cdots \subseteq L_r = [n]$, where labels in $L_1$ are simple functions of the input, while for $i > 1$, labels in $L_i$ are simple functions of simpler labels.

Our class surpasses models that were previously shown to be learnable by deep learning algorithms, in the sense that it reaches the depth limit of efficient learnability. That is, there are models in this class that require polynomial depth to express, whereas previous models can be computed by log-depth circuits.

Furthermore, we suggest that learnability of such hierarchical models might eventually form a basis for understanding deep learning. Beyond their natural fit for domains where deep learning excels, we argue that the mere existence of human "teachers" supports the hypothesis that hierarchical structures are inherently available. By providing granular labels, teachers effectively reveal "hints" or "snippets" of the internal algorithms used by the brain. We formalize this intuition, showing that in a simplified model where a teacher is partially aware of their internal logic, a hierarchical structure emerges that facilitates efficient learnability.

## 1. Introduction

A central objective in deep learning theory is to demonstrate that gradient-based algorithms can efficiently learn a class of models sufficiently rich to capture reality. This effort began over a decade ago, coincidental with the undeniable empirical success of deep learning. Initial theoretical results demonstrated that deep learning algorithms can learn linear models, followed later by proofs for simple non-linear models.

This progress is remarkable, especially considering that until recently, no models were known to be provably learnable by deep learning algorithms. Moreover, the field was previously dominated by hardness results indicating severe limitations on the capabilities of neural networks. However, despite this progress, learning linear or simple non-linear models is insufficient to explain the practical success of deep learning.

In this paper, we advance this research effort by showing that deep learning algorithms—specifically layerwise SGD on residual networks—provably learn *hierarchical models*. We consider a supervised learning setting with $n$ possible labels, where each example is associated with a subset of these labels. Let $\mathbf{f}^* \colon \mathcal{X} \to \{\pm 1\}^n$ be the ground truth labeling function. We assume an *unknown* hierarchy of labels $L_1 \subseteq L_2 \subseteq \cdots \subseteq L_r = [n]$ such that labels in $L_1$ are simple functions (specifically, polynomial thresholds) of the input, while for $i > 1$, any label in $L_i$ is a simple function of simpler labels (i.e., those in $L_{i-1}$).

We suggest that the learnability of hierarchical models offers a compelling basis for understanding deep learning. First, hierarchical models are natural in domains where neural networks excel. In computer vision, for instance, a first-level label might be "this pixel is red" (i.e. the input itself); a second-level label might be "curved line" or "dark region"; and a third-level label might be "leaf" or "rectangle", and so on. Similar hierarchies exist in text and speech processing. Indeed, this hierarchical structure motivated the development of successful architectures such as convolutional and residual networks.

Second, one might even argue further that the mere existence of human "teachers" supports the hypothesis that hierarchical labeling exists and can be supplied to the algorithm. Consider the classic problem of recognizing a car in an image. Early AI approaches (circa 1970s–80s) failed because they attempted to manually codify the cognitive algorithms used by the human brain. This was superseded by machine learning, which approximates functions based on input-output pairs. While this data-driven approach has surpassed human performance, the standard narrative of its

---

[1]Google Research and The Hebrew University. Correspondence to: Amit Daniely <amit.daniely@mail.huji.ac.il>.

*Proceedings of the $43^{rd}$ International Conference on Machine Learning*, Seoul, South Korea. PMLR 306, 2026. Copyright 2026 by the author(s).

success might be somewhat misleading.

We suggest that recent breakthroughs in deep learning stem not solely from "learning from scratch," but also from training on datasets rich in granular labels. (In modern LLMs, for instance, the vocabulary size exceeds $10^5$ tokens; considering pairs of tokens as labels expands this to the order of $10^{10}$ or $10^{11}$.) These labels represent a middle ground between explicit programming and pure input-output learning; they serve as "hints" or intermediate steps for learning complex concepts. Going back to the car recognition example, although we lack full access to the brain's internal algorithm, we can provide "snippets" of its logic. By supplying lower-level labels—such as windows, wheels, or geometric shapes—we effectively decompose the task into a hierarchy.

At a larger scale, we can consider the following perspective for the creation of LLMs. From the 1990s to the present, humanity created the internet (websites, forums, images, videos, etc.). As a byproduct, humanity implicitly provided an extensive number of labels and examples. Because these labels are so numerous—ranging from the very simple to the very complex—they are likely to possess a hierarchical structure. Following the creation of the internet, huge models were trained on these examples, succeeding largely as a result of this structure (alongside, of course, the extensive data volume and compute power). In a sense, the evolution of the internet and modern LLMs can be viewed as an enormous collective effort to create a circuit that mimics the human brain, in the sense that all labels of interest are effectively a composition of this circuit and a simple function.

We present a simplified formalization of this intuition. We model the human brain as a computational circuit, where each label (representing a "brain snippet") corresponds to a majority vote over a subset of the brain's neurons. To formalize the postulate that these labels are both granular and diverse, we assume that the specific collections of neurons defining each label are chosen at random prior to the learning process. We demonstrate that this setting yields a hierarchical structure that facilitates efficient learnability by residual networks. Crucially, neither the residual network architecture nor the training algorithm relies on knowledge of this underlying label hierarchy.

Finally, we note that hierarchical models surpass previous classes of models shown to be learnable by SGD. To the best of our knowledge, prior results were limited to models that can be realized by log-depth circuits. In contrast, hierarchical models *reach the depth limit of efficient learnability*. For any polynomial-sized circuit, we can construct a corresponding hierarchical model learnable by SGD on a ResNet, effectively computing the circuit as one of its labels.

**Related Work** Linear, or fixed representation models are defined by a fixed (usually non-linear) feature mapping followed by a learned linear mapping. This includes kernel methods, random features (Rahimi & Recht, 2007), and others. Several papers in the last decade have shown that neural networks can provably learn various linear models, e.g. (Andoni et al., 2014; Daniely et al., 2016; Du et al., 2017; Daniely, 2017; Li & Liang, 2018; Cao & Gu, 2019; Allen-Zhu et al., 2019; Daniely, 2020a;b). Several works consider model-classes which go beyond fixed representations, but still can be efficiently learned by gradient based methods on neural networks. One line of work shows learnability of parities under non-uniform distributions, or other models directly expressible by neural networks of depth two, e.g. (Livni et al., 2014; Ge et al., 2017; Tian, 2017; Frei et al., 2020; Daniely & Malach, 2020; Yehudai & Shamir, 2020; Vardi et al., 2021; Bietti et al., 2022; Bruna et al., 2023; Cornacchia & Mossel, 2023). Closer to our approach are (Abbe et al., 2021; Allen-Zhu & Li, 2020; Daniely et al., 2024; Wang et al., 2025) that consider certain hierarchical models. As mentioned above, we believe that our work is another step towards models that can capture reality. From a more formal perspective, we improve over previous work in the sense that the models we consider can be *arbitrarily deep*. In contrast, all the mentioned papers consider models that can be realized by networks of logarithmic depth. In fact, with the exception of (Wang et al., 2025) which considers composition of permutations, depth two suffices to express all the above mentioned models.

Another line of related work is (Li & Mossel, 2025; Huang & Mossel, 2025; Mossel, 2016; Koehler & Mossel, 2022) which argue that deep learning is successful due to hierarchical structure. This series of papers gives an example to a hierarchical model that is efficiently learnable, but it is conjectured that it requires deep architecture to express. Additional attempts to argue that hierarchy is essential for deep learning include (Patel et al., 2016; Mhaskar et al., 2017; Bruna & Mallat, 2013; Poggio & Fraser, 2024)

## 2. Notation and Preliminaries

We denote vectors using bold letters (e.g., $\mathbf{x}, \mathbf{y}, \mathbf{z}, \mathbf{w}, \mathbf{v}$) and their coordinates using standard letters. For instance, $x_i$ denotes the $i$-th coordinate of $\mathbf{x}$. Likewise, we denote vector-valued functions and polynomials (i.e., those whose range is $\mathbb{R}^d$) using bold letters (e.g., $\mathbf{f}, \mathbf{g}, \mathbf{h}, \mathbf{p}, \mathbf{q}, \mathbf{r}$), and their $i$-th coordinate using standard letters. We will freely use broadcasting operations. For instance, if $\vec{\mathbf{x}} = (\mathbf{x}_1, \ldots, \mathbf{x}_n)$ is a sequence $n$ of vectors in $\mathbb{R}^d$ and $g$ is a function from $\mathbb{R}^d$ to some set $Y$, then $g(\vec{\mathbf{x}})$ denotes the sequence $(g(\mathbf{x}_1), \ldots, g(\mathbf{x}_n))$. Similarly, for a matrix $A \in M_{q,d}$, we denote $A\vec{\mathbf{x}} = (A\mathbf{x}_1, \ldots, A\mathbf{x}_n)$.

For a polynomial $p : \mathbb{R}^n \to \mathbb{R}$, we denote by $\|p\|_{\mathrm{co}}$ the

Euclidean norm of the coefficient vector of $p$. We call $\|p\|_{\mathrm{co}}$ the *coefficient norm* of $p$. For $\sigma : \mathbb{R} \to \mathbb{R}$, we denote by $\|\sigma\| = \sqrt{\mathbb{E}_{X \sim \mathcal{N}(0,1)}[\sigma^2(X)]}$ the $\ell^2$ norm with respect to the standard Gaussian measure. We denote the Frobenius norm of a matrix $A \in M_{n,m}$ by $\|A\|_F = \sqrt{\sum_{i,j} A_{ij}^2}$, and the spectral norm by $\|A\| = \max_{\|\mathbf{x}\|=1} \|A\mathbf{x}\|$.

We denote by $\mathbb{R}^{d,n}$ the space of sequences of $n$ vectors in $\mathbb{R}^d$. More generally, for a set $G$, we let $\mathbb{R}^{d,G} = \{\vec{\mathbf{x}} = (\mathbf{x}_g)_{g \in G} : \forall g \in G, \ \mathbf{x}_g \in \mathbb{R}^d\}$. We denote the Euclidean unit ball by $\mathbb{B}^d = \{\mathbf{x} \in \mathbb{R}^d : \|\mathbf{x}\| \leq 1\}$. We denote the point-wise (Hadamard) multiplication of vectors and matrices by $\odot$ and the concatenation of vectors by $(\mathbf{x}|\mathbf{y})$. For $\mathbf{x} \in \mathbb{R}^n$, $A \subseteq [n]$, and $\sigma \in \mathbb{Z}^n$, we use the multi-index notation $\mathbf{x}^A = \prod_{i \in A} x_i$ and $\mathbf{x}^\sigma = \prod_{i=1}^n x_i^{\sigma_i}$. For $\mathbf{f} : \mathcal{X} \to \mathbb{R}^n$ and $L \subseteq [n]$, we denote by $\mathbf{f}_L : \mathcal{X} \to \mathbb{R}^{|L|}$ the restriction $\mathbf{f}_L = (f_{i_1}, \ldots, f_{i_k})$, where $L = \{i_1, \ldots, i_k\}$ with $i_1 < \ldots < i_k$. More generally, for $\mathbf{f} = (\mathbf{f}_i)_{i \in [n]} : \mathcal{X} \to \mathbb{R}^{n,G}$, we denote by $\mathbf{f}_L : \mathcal{X} \to \mathbb{R}^{|L|,G}$ the restriction $\mathbf{f}_L = (\mathbf{f}_{i_1}, \ldots, \mathbf{f}_{i_k})$.

## 2.1. Polynomial Threshold Functions

Fix a set $\mathcal{X} \subseteq [-1,1]^d$, a function $f : \mathcal{X} \to \{\pm 1\}$, a positive integer $K$, and $M > 0$. We say that $f$ is a $(K, M)$-PTF if there is a degree $\leq K$ polynomial $p : \mathbb{R}^d \to \mathbb{R}$ such that $\|p\|_{\mathrm{co}} \leq M$ and $\forall \mathbf{x} \in \mathcal{X}, \ p(\mathbf{x})f(\mathbf{x}) \geq 1$. More generally, we say that $f$ is a $(K, M)$-PTF of $\mathbf{h} : \mathcal{X} \to \mathbb{R}^s$ if there is a degree $\leq K$ polynomial $p : \mathbb{R}^s \to \mathbb{R}$ such that $\|p\|_{\mathrm{co}} \leq M$ and $\forall \mathbf{x} \in \mathcal{X}, \ p(\mathbf{h}(\mathbf{x}))f(\mathbf{x}) \geq 1$. An example of a $(K, 1)$-PTF that we will use frequently is a function $f : \{\pm 1\}^d \to \{\pm 1\}$ that depends on $K$ variables. Indeed, Fourier analysis on $\{\pm 1\}^d$ tells us that $f$ is a restriction of a degree $\leq K$ polynomial $p$ with $\|p\|_{\mathrm{co}} = 1$. For this polynomial we have $\forall \mathbf{x} \in \mathcal{X}, \ p(\mathbf{x})f(\mathbf{x}) = 1$.

We will also need a more refined definitions of PTFs, which allows us to require two sided inequality $B \geq p(\mathbf{x})f(\mathbf{x}) \geq 1$, as well as some robustness to perturbation of $\mathbf{x}$. To this end, for $\mathbf{x} \in [-1,1]^d$ and $r > 0$ we define

$$\mathcal{B}_r(\mathbf{x}) = \left\{ \tilde{\mathbf{x}} \in [-1,1]^d : \|\mathbf{x} - \tilde{\mathbf{x}}\|_\infty \leq r \right\} \quad (1)$$

Fix $B \geq 1$ and $1 \geq \xi > 0$. We say that $f$ is a $(K, M, B, \xi)$-PTF if there is a degree $\leq K$ polynomial $p : \mathbb{R}^d \to \mathbb{R}$ such that $\|p\|_{\mathrm{co}} \leq M$ and

$$\forall \mathbf{x} \in \mathcal{X} \ \forall \tilde{\mathbf{x}} \in \mathcal{B}_\xi(\mathbf{x}), \ \ B \geq p(\tilde{\mathbf{x}})f(\mathbf{x}) \geq 1$$

Likewise, we say that $f$ is a $(K, M, B, \xi)$-PTF of $\mathbf{h} = (h_1, \ldots, h_s) : \mathcal{X} \to [-1,1]$ if there is a degree $\leq K$ polynomial $p : \mathbb{R}^s \to \mathbb{R}$ such that $\|p\|_{\mathrm{co}} \leq M$ and

$$\forall \mathbf{x} \in \mathcal{X} \ \forall \mathbf{y} \in \mathcal{B}_\xi(\mathbf{h}(\mathbf{x})), \ \ B \geq p(\mathbf{y})f(\mathbf{x}) \geq 1$$

Finally, we say that $f$ is a $(K, M, B)$-PTF (resp. $(K, M, B)$-PTF of $\mathbf{h}$) if it is a $(K, M, B, 1)$-PTF (resp. $(K, M, B, 1)$-PTF of $\mathbf{h}$).

## 2.2. Strong Convexity

Let $W \subseteq \mathbb{R}^d$ be convex. We say that a differentiable $f : W \to \mathbb{R}$ is $\lambda$-strongly-convex if for any $\mathbf{x}, \mathbf{y} \in W$ we have

$$f(\mathbf{y}) \geq f(\mathbf{x}) + \langle \mathbf{y} - \mathbf{x}, \nabla f(\mathbf{x}) \rangle + \frac{\lambda}{2} \|\mathbf{y} - \mathbf{x}\|^2$$

We note that if $f$ is strongly convex and $\|\nabla f(\mathbf{x})\| \leq \epsilon$ for $\mathbf{x} \in W$ then $\mathbf{x}$ minimizes $f$ up to an additive error of $\frac{\epsilon^2}{2\lambda}$. Indeed, for any $\mathbf{y} \in W$ we have

$$
\begin{aligned}
f(\mathbf{x}) & \leq & f(\mathbf{y}) - \frac{\lambda}{2} \|\mathbf{y} - \mathbf{x}\|^2 + \|\mathbf{y} - \mathbf{x}\| \cdot \|\nabla f(\mathbf{x})\| \\
& = & f(\mathbf{y}) + \frac{\|\nabla f(\mathbf{x})\|^2}{2\lambda} \\
& & - \frac{1}{2\lambda} \left( \|\nabla f(\mathbf{x})\| - \lambda \|\mathbf{y} - \mathbf{x}\| \right)^2 \\
& \leq & f(\mathbf{y}) + \frac{\|\nabla f(\mathbf{x})\|^2}{2\lambda} \\
& \leq & f(\mathbf{y}) + \frac{\epsilon^2}{2\lambda}
\end{aligned}
\quad (2)
$$

## 2.3. Hermite Polynomials

The results we state next can be found in (Andrews et al., 1999). The Hermite polynomials $h_0, h_1, h_2, \ldots$ are the sequence of orthonormal polynomials corresponding to the standard Gaussian measure $\mu$ on $\mathbb{R}$. That is, they are the sequence of orthonormal polynomials obtained by the Gram-Schmidt process of $1, x, x^2, x^3, \ldots \in L^2(\mu)$. The Hermite polynomials satisfy the following recurrence relation

$$x h_n(x) = \sqrt{n+1} h_{n+1}(x) + \sqrt{n} h_{n-1}(x) \quad (3)$$

and $h_0(x) = 1, \ h_1(x) = x$ or equivalently

$$h_{n+1}(x) = \frac{x}{\sqrt{n+1}} h_n(x) - \sqrt{\frac{n}{n+1}} h_{n-1}(x)$$

The generating function of the Hermite polynomials is

$$e^{xt - \frac{t^2}{2}} = \sum_{n=0}^\infty \frac{h_n(x) t^n}{\sqrt{n!}} \quad (4)$$

We also have

$$h_n' = \sqrt{n} h_{n-1} \quad (5)$$

Likewise, if $X, Y \sim \mathcal{N}\left(0, \begin{pmatrix} 1 & \rho \\ \rho & 1 \end{pmatrix}\right)$

$$\mathbb{E} h_i(X) h_j(Y) = \delta_{ij} \rho^i \quad (6)$$

## 3. The Hierarchical Model

Let $\mathcal{X} \subseteq [-1,1]^d$ be our instance space. We consider the multi-label setting, in which each instance can have anything

between 0 to $n$ positive labels, and each training example comes with a list of all[1] its positive labels. Hence, our goal is to learn the labeling function $\mathbf{f}^* : \mathcal{X} \to \{\pm 1\}^n$ based on a sample

$$S = \{(\mathbf{x}^1, \mathbf{f}^*(\mathbf{x}^1)), \dots, (\mathbf{x}^m, \mathbf{f}^*(\mathbf{x}^m))\} \in (\mathcal{X} \times \{\pm 1\}^n)^m$$

of i.i.d. labeled examples that comes from a distribution $\mathcal{D}$ on $\mathcal{X}$. Specifically, our goal is to find a predictor $\hat{\mathbf{f}} : \mathcal{X} \to \mathbb{R}^n$ whose error, $\mathrm{Err}_{\mathcal{D}}(\hat{\mathbf{f}}) = \mathrm{Pr}_{\mathbf{x} \sim \mathcal{D}} \left( \mathrm{sign}(\hat{\mathbf{f}}(\mathbf{x})) \neq \mathbf{f}^*(\mathbf{x}) \right)$, is small. We assume that there is a hierarchy of labels (unknown to the algorithm), with the convention that

- The first level of the hierarchy consists of labels which are simple (= easy to learn) functions of the input. Specifically, each such label is a polynomial threshold function (PTF) of the input.

- Any label in the $i$'th level of the hierarchy is a simple function (again, a PTF) of labels from lower levels of the hierarchy.

We next give the formal definition of hierarchy.

**Definition 3.1** (hierarchy). Let $\mathcal{L} = \{L_1, \dots, L_r\}$ be a collection of sets such that $L_1 \subseteq L_2 \subset \dots \subseteq L_r = [n]$. We say that $\mathcal{L}$ is a hierarchy for $\mathbf{f}^* : \mathcal{X} \to \{\pm 1\}^n$ of complexity $(r, K, M)$ (or $(r, K, M)$-hierarchy for short) if for any $j \in L_1$ the function $f_j^*$ is a $(K, M)$-PTF and for $i \geq 2$, and $j \in L_i$ we have that $\mathbf{f}_j^* = \tilde{f}_j \circ \mathbf{f}_{L_{i-1}}^*$ for a $(K, M)$-PTF $\tilde{f}_j : \{\pm 1\}^{|L_{i-1}|} \to \{\pm 1\}$.

*Example* 3.2. Fix $\mathcal{L} = \{L_1, \dots, L_r\}$ as in Definition 3.1, and recall that a boolean function that depends on $K$ coordinates is a $(K, 1)$-PTF. Hence, if for any $i \geq 2$, any label $j \in L_i$ depends on at most $K$ labels from $L_{i-1}$, and any label $j \in L_1$ is a $(K, 1)$-PTF of the input, then $\mathcal{L}$ is an $(r, K, 1)$-hierarchy.

Assuming that $K$ is constant, our main result will show that given $\mathrm{poly}(n, d, M, 1/\epsilon)$ samples, a poly-time SGD algorithm on a residual network of size $\mathrm{poly}(n, d, M, 1/\epsilon)$ can learn any function $\mathbf{f}^* : \mathcal{X} \to \{\pm 1\}^n$ with error of $\epsilon$, provided that $\mathbf{f}^*$ has a hierarchy of complexity $(r, K, M)$ (the algorithm and the network do not depend on the hierarchy, but just on $r, K, M$).

One of the steps in the proof of this result is to show that any $(K, M)$-PTF on a subset of $[-1, 1]^n$ is necessarily a $(K, 2M, B, \xi)$-PTF for $\xi = \frac{1}{2(n+1)^{\frac{K+1}{2}} KM}$ and

---

[1] We note that in practice it is often the case that an example possesses several positive labels (for instance, "dog" and "animal"). However, each training example usually comes with just one of its positive labels. We hope that future work will be able to handle this more realistic type of supervision.

$B = 2(\max(n, d) + 1)^{K/2} M$ (see Lemma C.2). This is enough for establishing our main result as informally described above. Yet, in some cases of interest, we can have much larger $\xi$ and smaller $B$. In this case, we can guarantee learnability with smaller network, and less samples and runtime. Hence, we next refine the definition of hierarchy by adding $B$ and $\xi$ as parameters.

**Definition 3.3** (hierarchy). Let $\mathcal{L} = \{L_1, \dots, L_r\}$ be a collection of sets such that $L_1 \subseteq L_2 \subset \dots \subseteq L_r = [n]$. We say that $\mathcal{L}$ is a hierarchy for $\mathbf{f}^* : \mathcal{X} \to \{\pm 1\}^n$ of complexity $(r, K, M, B, \xi)$ (or $(r, K, M, B, \xi)$-hierarchy for short) if for any $j \in L_1$ the function $f_j^*$ is a $(K, M, B)$-PTF and for $i \geq 2$, and $j \in L_i$ we have that $\mathbf{f}_j^* = \tilde{f}_j \circ \mathbf{f}_{L_{i-1}}^*$ for a $(K, M, B, \xi)$-PTF $\tilde{f}_j : \{\pm 1\}^{|L_{i-1}|} \to \{\pm 1\}$.

### 3.1. The "Brain Dump" Hierarchy

Fix a domain $\mathcal{X} \subseteq \{\pm 1\}^d$ and a sequence of functions $G^i : \{\pm 1\}^d \to \{\pm 1\}^d$ for $1 \leq i \leq r$. We assume that $G^0(\mathbf{x}) = \mathbf{x}$, and for any depth $i \in [r]$ and coordinate $j \in [d]$, we have

$$\forall \mathbf{x} \in \mathcal{X}, \quad G_j^i(\mathbf{x}) = h_j^i(G^{i-1}(\mathbf{x})),$$

where $h_j^i : \{\pm 1\}^d \to \{\pm 1\}$ is a function that depends on $K$ coordinates. We view the sequence $G^1, \dots, G^r$ as a computation circuit, or a model of a "brain."

Suppose we wish to learn a function of the form $f^* = h \circ G^r$, where $h : \{\pm 1\}^d \to \{\pm 1\}$ also depends only on $K$ inputs, given access to labeled samples $(\mathbf{x}, f^*(\mathbf{x}))$. The function $f^*$ can be extremely complex. For instance, $G$ could compute a cryptographic function. In such cases, learning $f^*$ solely from labeled examples $(\mathbf{x}, f^*(\mathbf{x}))$ is likely intractable; if our access to $f^*$ is restricted to the black-box scenario described above, the task appears impossible. On the other extreme, if we had complete white-box access to $f^*$—meaning a full description of the circuit $G$—the learning problem would become trivial. However, if $G$ truly models a human brain, such transparent access is unrealistic.

Consider a middle ground between these black-box and white-box scenarios. Assume we can query the labeler (the human whose brain is modeled by $G$) for additional information. For instance, if $f^*$ is a function that recognizes cars in an image, we can ask the labeler not only whether the image contains a car, but also to identify specific features: wheels, windows, dark areas, curves, and whatever *he thinks* is relevant. Each of these additional labels represents another simple function computed over the circuit $G$. We model these auxiliary labels as random majorities of randomly chosen $G_j^i$'s. We show that with enough such labels, the resulting problem admits a low-complexity hierarchy and is therefore efficiently learnable.

Formally, fix an integer $q$. We assume that for every depth

$i \in [r]$, there are $q$ auxiliary labels $f_{i,j}^*$ for $1 \leq j \leq q$, each of which is a signed Majority of an odd number of components of $G^i$. Moreover, we assume these functions are random. Specifically, prior to learning, the labeler independently samples $qr$ functions such that for any $i \in [r]$ and $j \in [q]$,

$$f_{i,j}^*(\mathbf{x}) = \text{sign}\left(\sum_{l=1}^{d} w_l^{i,j} G_l^i(\mathbf{x})\right),$$

where the weight vectors $\mathbf{w}^{i,j} \in \mathbb{R}^d$ are independent uniform vectors chosen from

$$\mathcal{W}_{d,k} := \left\{\mathbf{w} \in \{-1,0,1\}^d : \sum_{l=1}^{d} |w_l| = k\right\}$$

for some odd integer $k$.

**Theorem 3.4.** *If $q = \tilde{\omega}\left(k^2 d \log(|\mathcal{X}|)\right)$ then $\mathbf{f}^*$ has $\left(r, K, O\left(kd^K\right), 2k+1\right)$-hierarchy w.p. $1 - o(1)$*

### 3.2. Extension to Sequential and Ensemble Models

We next extend the notion of hierarchy for the common setting in which the input and the output of the learned function is an ensemble of vectors. Let $G$ be some set. We will refer to elements in $G$ as *locations*. In the context of images a natural choice would be $G = [T_1] \times [T_2]$, where $T_1 \times T_2$ is the maximal size of an input image. In the context of language a natural choice would be $G = [T]$, where $T$ is the maximal number of tokens in the input. We denote by $\vec{x} = (\mathbf{x}_g)_{g \in G}$ ensemble of vectors and let $\mathbb{R}^{d,G} = \{\vec{x} = (\mathbf{x}_g)_{g \in G} : \forall g \in G, \ \mathbf{x}_g \in \mathbb{R}^d\}$.

Fix $\mathcal{X} \subseteq [-1,1]^d$ and let $\mathcal{X}^G$ be our instance space. Assume that there are $n$ labels. We consider the setting in which each instance at each location can have anything between $0$ to $n$ positive labels. In light of that, our goal is to learn the labeling function $\mathbf{f}^* : \mathcal{X}^G \to \{\pm 1\}^{n,G}$ based on a sample

$$S = \{(\vec{x}^t, \mathbf{f}^*(\vec{x}^t))\}_{t=1}^{m} \in \left(\mathcal{X}^G \times \{\pm 1\}^{n,G}\right)^m$$

of i.i.d. labeled examples coming from a distribution $\mathcal{D}$ on $\mathcal{X}^G$. We assume that there is a hierarchy of labels (unknown to the algorithm), with the convention that

- The first level of the hierarchy consists of labels which are simple (= easy to learn) functions of the input. Specifically, each such label at location $g$ is a PTF of the input *near $g$*.

- Any label in the $i$'th level of the hierarchy is a simple function of labels from lower levels. Specifically, each such label at location $g$ is a PTF of lower level labels, at locations near $g$.

We will capture the notion of proximity of locations in $G$ via a *proximity mapping*, which designates $w$ nearby locations to any element $g \in G$. We will always consider $g$ itself as a point near $g$. This is captured in the following definition

**Definition 3.5** (proximity mapping). A *proximity mapping* of *width $w$* is a mapping $\mathbf{e} = (e_1, \ldots, e_w) : G \to G^w$ such that $e_1(g) = g$ for any $g$.

For instance, if $G = [T]$, it is natural to choose $\mathbf{e} : G \to G^{2w+1}$ such that $\{e_1(g), \ldots, e_{2w+1}(g)\} = \{g' \in T : |g' - g| \leq w\}$. Likewise, if $G = [T] \times [T]$, it is natural to choose $\mathbf{e} : G \to G^{(2w+1)^2}$ such that $\{e_1(g_1, g_2), \ldots, e_{(2w+1)^2}(g_1, g_2)\} = \{(g_1', g_2') \in T \times T : |g_1' - g_1| \leq w \text{ and } |g_2' - g_2| \leq w\}$. Given a proximity mapping $\mathbf{e}$ and $\vec{x} \in \mathbb{R}^{d,G}$ we define $E_g(\vec{x})$ as the concatenation of all vectors $\mathbf{x}_{g'}$ where $g'$ is close to $g$ according to $\mathbf{e}$. Formally,

**Definition 3.6.** Given a proximity mapping $\mathbf{e} : G \to G^w$, $g \in G$ and $\vec{x} \in \mathbb{R}^{d,G}$ we define $E_g(\vec{x}) = (\mathbf{x}_{e_1(g)}|\ldots|\mathbf{x}_{e_w(g)}) \in \mathbb{R}^{dw}$. Likewise, we let $E(\vec{x}) \in \mathbb{R}^{dw,G}$ be $E(\vec{x}) = (E_g(\vec{x}))_{g \in G}$.

We next extend the definition of PTF to accommodate the ensemble setting.

**Definition 3.7** (hierarchy). Let $\mathcal{L} = \{L_1, \ldots, L_r\}$ be a collection of sets such that $L_1 \subseteq L_2 \subset \ldots \subseteq L_r = [n]$. Let $\mathbf{e} : G \to G^w$ be a proximity function. We say that $(\mathcal{L}, \mathbf{e})$ is a hierarchy for $\mathbf{f}^* : \mathcal{X}^G \to \{\pm 1\}^{n,G}$ of complexity $(r, K, M, B, \xi)$ (or $(r, K, M, B, \xi)$-hierarchy for short) if

- For any $j \in L_1$ there is a $(K, M, B, \xi)$-PTF $\tilde{f}_j : \mathcal{X}^w \to \{\pm 1\}$ such that $f_{j,g}(\mathbf{x}) = \tilde{f}(E_g(\mathbf{x}))$ for any $\mathbf{x} \in \mathcal{X}^G$ and $g \in G$

- For $i \geq 2$, and $j \in L_i$ there is a $(K, M, B, \xi)$-PTF $\tilde{f}_j : \{\pm 1\}^{|L_1|w} \to \{\pm 1\}$ such that $f_{j,g}(\mathbf{x}) = \tilde{f}(E_g(\mathbf{f}_{L_{i-1}}^*(\mathbf{x})))$ for any $\mathbf{x} \in \mathcal{X}^G$ and $g \in G$

We note that the previous definition of hierarchy (i.e. definitions 3.1 and 3.3) is the special case $w = |G| = 1$.

## 4. Algorithm and Main Result

Fix $\mathcal{X} \subseteq [-1,1]^d$, a location set $G$, a proximity mapping $e : G \times N \to G$ of width $w$, some constant integer $K \geq 1$, and an activation function $\sigma : \mathbb{R} \to \mathbb{R}$ that is Lipschitz, bounded and is not a constant function. We will view $\sigma$ and $K$ as fixed, and will allow big-$O$ notation to hide constants that depend on $\sigma$ and $K$.

We start by describing the residual network architecture that we will consider. Let $\mathcal{X}^G$ be our instance space. The first layer (actually, it is two layers, but it will be easier to

---

**Algorithm 1** Resnet Training

---

At each step $k = 1, \ldots, D - 1$ optimize the $\ell_S(\vec{W}) + \frac{\epsilon_{\mathrm{opt}}}{2}\|W_2^k\|^2$ over $W_2^k$, until a gradient of size $\leq \epsilon_{\mathrm{opt}}$ is reached. (as the $k$'th step objective is $\epsilon_{\mathrm{opt}}$-strongly convex the algorithm finds an $\frac{\epsilon_{\mathrm{opt}}}{2}$-minimizer of it.)

---

consider it as one layer) of the network will compute the function

$$\Psi_1(\vec{\mathbf{x}}) = W_2^1 \sigma(W_1^1 E(\vec{\mathbf{x}}) + \mathbf{b}^1)$$

We assume that $W_2^1 \in \mathbb{R}^{n \times q}$ is initialized to 0, while $(W_1^1, \mathbf{b}^1) \in \mathbb{R}^{q \times wd} \times \mathbb{R}^q$ is initialized using $\beta$-*Xavier initialization* as defined next.

**Definition 4.1** (Xavier Initialization). Fix $1 \geq \beta \geq 0$. A random pair $(W, \mathbf{b}) \in \mathbb{R}^{q \times d} \times \mathbb{R}^q$ has $\beta$-*Xavier distribution* if the entries of $W$ are i.i.d. centered Gaussians of variance $\frac{1-\beta^2}{d}$, and $\mathbf{b}$ is independent from $W$ and its entries are i.i.d. centered Gaussians of variance $\beta^2$

The remaining layers are of the form

$$\Psi_k(\vec{\mathbf{x}}) = \vec{\mathbf{x}} + W_2^k \sigma(W_1^k E(\vec{\mathbf{x}}) + \mathbf{b}^k)$$

where $(W_1^k, \mathbf{b}^k) \in \mathbb{R}^{q \times (wn)} \times \mathbb{R}^q$ is initialized using $\beta$-Xavier initialization and $W_2^k \in \mathbb{R}^{n \times q}$ is initialized to 0. Finally, the last layer computes

$$\Psi_D(\vec{\mathbf{x}}) = W^D \vec{\mathbf{x}}$$

for an orthogonal matrix $W^D \in \mathbb{R}^{n \times n}$. We will denote the collection of weight matrices by $\vec{W}$, and the function computed by the network by $\hat{\mathbf{f}}_{\vec{W}}$. Fix a convex loss function $\ell : \mathbb{R} \to [0, \infty)$ we extend it to a loss $\ell : \mathbb{R}^G \times \{\pm 1\}^G \to [0, \infty)$ by averaging:

$$\ell(\hat{\mathbf{y}}, \mathbf{y}) = \frac{1}{|G|} \sum_{g \in G} \ell(\hat{y}_g \cdot y_g)$$

Likewise, for a function $\hat{\mathbf{f}} : \mathcal{X}^G \to \mathbb{R}^{n,G}$ and $j \in [n]$ we define

$$\ell_{S,j}(\hat{\mathbf{f}}) = \ell_{S,j}(\hat{\mathbf{f}}_j) = \frac{1}{m} \sum_{t=1}^m \ell\left(\hat{\mathbf{f}}_j(\vec{\mathbf{x}}^t), \mathbf{y}_j^t\right)$$

Finally, let

$$\ell_S(\hat{\mathbf{f}}) = \sum_{j=1}^n \ell_{S,j}(\hat{\mathbf{f}}) \quad \text{and} \quad \ell_S(\vec{W}) = \ell_S\left(\hat{\mathbf{f}}_{\vec{W}}\right)$$

We will consider the following algorithm

We will consider the following loss function.

$$\ell = \ell_{1/(2B)} + \frac{1}{4m|G|}\ell_{1-\xi/2} \tag{7}$$

for

$$\ell_\eta(z) = \begin{cases} 1 - \frac{z}{\eta} & 0 \leq z \leq \eta \\ 0 & \eta \leq z \leq 1 \\ \infty & \text{otherwise} \end{cases}$$

We are now ready to state our main result.

**Theorem 4.2** (Main). *Assume that $\mathbf{f}^*$ has $(r, K, M, B, \xi)$-hierarchy and let $\gamma = \frac{1}{32}\min\left(\frac{1}{B}, \xi\right)$. Assume that*

- $D > r \cdot \left(\left\lceil \frac{\ln(8m|G|/\xi)}{\gamma} \right\rceil + 1\right)$

- $\epsilon_{\mathrm{opt}} \leq \frac{(1-e^{-\gamma})\xi}{16m^2|G|^2}$

*Then, there is a choice of $\beta$ and $q = \tilde{O}\left(\frac{(M+1)^4(wn)^{2K}}{\gamma^{4+2K}}\right)$ such that algorithm 1 will learn a classifier with expected error at most $\tilde{O}\left(\frac{D^2(M+1)^4(wn)^{2K+1}}{\gamma^{4+2K}m}\right)$.*

## 5. Proof Sketch of Main Result

The proof proceeds by induction on the depth of the hierarchy $r$. We demonstrate that if the network has successfully learned the labels in level $i - 1$ of the hierarchy, training subsequent layers via Stochastic Gradient Descent (SGD) will efficiently learn the labels in level $i$.

**1. Recursive Structure and Layerwise Training.** We assume the network is trained layer-by-layer. Let $\hat{\mathbf{f}}^{(k)}$ denote the network's output after training $k$ blocks. The induction hypothesis assumes that for some $k$, $\hat{\mathbf{f}}^{(k)}$ constitutes a sufficiently accurate approximation of the labels $L_{i-1}$ (the $(i - 1)$-th level of the hierarchy). We show that training a constant number of additional blocks allows the network to learn the labels $L_i$ with high probability.

**2. Robustness of Polynomial Threshold Functions.** A key technical challenge is that the inputs to the current layer are not the ground-truth labels from the previous level, but rather the network's approximations. To address this, we establish a robustness property for Polynomial Threshold Functions (PTFs). Specifically, Lemma C.2 (or Lemma **??**) shows that any $(K, M)$-PTF inherently possesses a "robust margin." If a label $y$ is defined as $y = \mathrm{sign}(p(\mathbf{z}))$, there exists a polynomial $P$ such that $y \cdot P(\tilde{\mathbf{z}}) \geq 1$ holds even for perturbed inputs $\tilde{\mathbf{z}}$, provided $\|\mathbf{z} - \tilde{\mathbf{z}}\|_\infty$ is sufficiently small. This ensures that approximation errors from previous layers do not prevent the current layer from learning the next level's labels.

**3. Kernel Approximation in Residual Blocks.** We analyze the learnability of each residual block using the theory of Random Features. We model each residual block (with

frozen internal weights and a trained readout) as a linear model over a feature map $\Psi$ induced by the activation function.

Using the Hermite polynomial expansion of the activation function, we prove that the Neural Tangent Kernel (or the specific Random Features kernel) associated with the block can approximate any degree-$K$ polynomial. Consequently, the robust polynomial $P$ required to classify the labels of $L_i$ (given inputs approximating $L_{i-1}$) lies within the Reproducing Kernel Hilbert Space (RKHS) of the network block and has a bounded norm.

**4. Convergence of Layerwise SGD.** Since layerwise training effectively solves a sequence of convex optimization problems (learning the last layer weights given fixed features), we apply standard generalization bounds for SGD on linear models. Combining the kernel approximation result with the robustness property, we guarantee that SGD converges to a solution with small classification error for the labels in $L_i$. Repeating this process $r$ times allows the network to sequentially learn the entire hierarchy, recovering the final target function $\mathbf{f}^*$.

# 6. Conclusion and Future Work

In this work, we argued that the availability of extensive and granular labeling suggests that the target functions in modern deep learning are inherently hierarchical, and we showed that deep learning—specifically, SGD on residual networks—can exploit such hierarchical structure. Our proof builds on a layerwise mechanism of the learning process, where each layer acts simultaneously as a representation learner and a predictor, iteratively refining the output of the previous layer. Our results give rise to several perspectives, which we outline below:

- **Supervised Learning is inherently tractable.** Contrary to worst-case hardness results, the existence of a teacher (and thus a hierarchy) implies that the problem is learnable in polynomial time, given the right supervision.

- **Very deep models are provably learnable.** Unlike previous theoretical works, we prove that ResNets can learn models that are realizable only by very deep circuits.

- **A middle ground between Software Engineering and Learning.** Modern deep learning can be viewed as a *relaxation of software engineering* and a *strengthening of classical learning*. Instead of manually "codifying the brain's algorithm" (traditional AI) or learning blindly from input-output pairs (classical ML), we provide snippets of the brain's logic via related labels.

This approach renders the learning task feasible without requiring full knowledge of the underlying circuit.

- **A modified narrative for learning theory.** Historically, the narrative governing learning theory, particularly from a computational perspective, has been the following: (i) Learning all functions is impossible. (ii) Upon closer inspection, we are interested only in functions that are efficiently computable. (iii) This function class is learnable using polynomial samples. (iv) Unfortunately, learning it requires exponential time. (v) Nevertheless, some simple function classes are learnable.

  The aforementioned narrative, however, is at odds with practice. Our work suggests that it might be possible to replace item (v) with the following: "(v) Re-evaluating our scope, we are primarily interested in functions that are efficiently computable *by humans*. (vi) We have good reasons to believe that these functions are hierarchical. (vii) As a result, they are learnable using polynomial time and samples."

Our work suggests using hierarchical models as a basis for understanding neural networks. Significant future work is required to advance this direction. First, theoretically, it would be useful to extend the scope of hierarchical models. To this end, one might:

- Analyze attention mechanisms through the lens of hierarchical models.

- Extend hierarchical models to capture a "single-function hierarchy." This refers to a scenario where a function $f$ possesses "simple versions" that are easy to acquire; crucially, mastering these simpler versions would render $f$ itself learnable. This direction aligns with prior work on the learnability of non-linear models via gradient-based algorithms (e.g., (Abbe et al., 2021)), which often (sometimes implicitly) assumes this structural hierarchy. More broadly, future models could be generalized to capture scenarios where access to *approximate versions* of labels facilitates the learning of some label.

- Extend the inherent justification of hierarchical models by generalizing Theorem 3.4. That is, define formal models of teachers that are "partially aware" to their internal logic, and show that hierarchical labeling which facilitates efficient learnability can be provided by such teachers. Put differently, show that "generic non-linear projection" of a hierarchical function is hierarchical itself.

- Identify low-complexity hierarchies for known algorithms. This could lead to new hierarchical architectures, and might even shed some light on how humans

discovered these algorithms, and facilitate teaching them.

Second, on the empirical side, it would be valuable to:

- Build practical learning algorithms with principled optimization procedures based more directly on the hierarchical learning perspective.

- Empirically test the hypothesis that, given enough labels, real-world data exhibits a hierarchical structure. In this respect, finding this explicit hierarchical structure can be viewed as an interpretation of the learned model.

Finally, we address specific limitations of our results, which rely on several assumptions. We outline the most prominent ones here, hoping that future work will be able to relax these constraints.

We begin with the technical assumptions. A clear direction for future work is to improve our quantitative bounds; while polynomial, they are likely far from optimal. Other technical constraints include the assumption that the output matrix is orthogonal and that the number of labels equals the dimension of the hidden layers. It would be more natural to consider an arbitrary number of labels and an output matrix initialized as a Xavier matrix (we note, however, that Xavier matrices are "almost orthogonal"). Finally, the loss function used in our analysis is non-standard.

Next, we address more inherent limitations. First, we assumed extremely strong supervision: that each example comes with all positive labels it possesses. In practice, one usually obtains only a single positive label per example. We note that while it is straightforward to show that hierarchical models are efficiently learnable with this standard supervision, proving that gradient-based algorithms on neural networks succeed in this setting remains an open problem.

Another limitation is our assumption of layer-wise training, whereas in reality, all layers are typically trained jointly. While this makes the mathematical analysis more intricate, joint training is likely superior for several reasons. First, empirically, it is the standard method. Second, if the goal of training lower layers is merely to learn representations, there is little utility in exhausting data to achieve marginal improvements in the loss. Indeed, to ensure data efficiency, it is preferable to utilize features as soon as they are sufficiently good (i.e., once the gradient w.r.t. these features is large).

## Impact Statement

This paper presents work whose goal is to advance the field of machine learning. There are many potential societal

consequences of our work, none of which we feel must be specifically highlighted here

## Acknowledgments

The research described in this paper was funded by the European Research Council (ERC) under the European Union's Horizon 2022 research and innovation program (grant agreement No. 101041711), and the Simons Foundation (as part of the Collaboration on the Mathematical and Scientific Foundations of Deep Learning). The author thanks Elchanan Mossel and Mariano Schain for useful comments.

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

## A. Proof of Theorem 4.2: Hierarchical Learning by Resnets

In order to prove Theorem 4.2 it is enough to prove Theorem A.1 below, which shows that there is a choice of $\beta$ and $q = \tilde{O}\left(\frac{(M+1)^4(wn)^{2K}}{\gamma^{4+2K}}\right)$ such that algorithm 1 will learn a classifier with empirical large margin error of 0 w.p. $\frac{1}{m}$. That is, we define

$$\text{Err}_{S,\gamma}(\hat{\mathbf{f}}) = \frac{1}{m}\sum_{t=1}^{m} 1\left[\exists(i,g)\in[n]\times G \text{ s.t. } \hat{f}_{i,g}(\vec{\mathbf{x}}^t)\cdot f_{i,g}^*(\vec{\mathbf{x}}^t) < \gamma\right] \tag{8}$$

And show that algorithm 1 will learn a classifier $\hat{\mathbf{f}}$ with $\text{Err}_{S,1/2}(\hat{\mathbf{f}}) = 0$ w.p. $\frac{1}{m}$. Let's call such an algorithm $(1/m)$-consistent. Given this guarantee, Theorem 4.2 will follow from a standard parameter counting argument: The number of trained parameters is $p = Dqn$, and their magnitude is bounded by $\frac{2n}{\epsilon_{\text{opt}}} + 1$ due to the $\ell^2$ regularization term. Likewise, excluding the small probability event that one of the initial weights has magnitude $\geq \ln(Dq(n+d)wm)$ (which happens w.p. $\ll \frac{1}{m}$, since all $Dq(n+d)w$ initial weights are centered Guassians with variance $\leq 1$), it is not hard to verify that as a composition of $2D$ layers, the network's output is $L$-Lipchitz w.r.t. the trained parameters for $L = 2^{\tilde{O}(D)}$. Thus, the expected error of any $(1/m)$-consistent algorithm is $\tilde{O}\left(\frac{p\log(L)}{m}\right) = \tilde{O}\left(\frac{Dp}{m}\right) = \tilde{O}\left(\frac{D^2qn}{m}\right)$. (See Lemma B.7 for a precise statement).

**Theorem A.1** (Main - Restated). *Let* $\gamma = \frac{1}{32}\min\left(\frac{1}{B},\xi\right)$. *Assume that*

- $\mathbf{f}^*$ *has* $(r,K,M,B,\xi)$-*hierarchy* $(\mathcal{L},e)$

- $D > r\cdot\left(\left\lceil\frac{\ln(8m|G|/\xi)}{\gamma}\right\rceil + 1\right)$

- $\epsilon_{\text{opt}} \leq \frac{(1-e^{-\gamma})\xi}{16m^2|G|^2}$

*There is a choice of* $\beta$ *such that w.p.* $1 - 2nmD|G|\exp\left(-\Omega\left(q\cdot\frac{\gamma^{2K+4}}{(wn)^{2K}(M+1)^4}\right)\right)$ *over the initial choice of the weights, Algorithm 1 will learn a classifier* $\hat{\mathbf{f}} : \mathcal{X}^G \to \mathbb{R}^{n,G}$ *with* $\text{Err}_{S,1/2}(\hat{\mathbf{f}}) = 0$.

For $1 \leq k \leq D$, let $\hat{\mathbf{f}}^k : \mathcal{X}^G \to \mathbb{R}^{n,G}$ be the function computed by the network after the $k$'th layer is trained. Also, let $\Gamma^k : \mathcal{X}^G \to \mathbb{R}^{n,G}$ be the function computed by the layers 1 to $k$ after the $k$'th layer is trained. For $k = 0$ we denote by $\hat{\mathbf{f}}^0 = \Gamma^0$ the identity mapping from : $\mathcal{X}^G$ to $\mathbb{R}^{d,G}$. We note that when algorithm 1 trains the $k$'th layer we have $W_2^{k'} = 0$ for any $k' > k$. Hence,

$$\Psi_{k'}(\vec{\mathbf{x}}) = \vec{\mathbf{x}} + W_2^{k'}\sigma(W_1^{k'}E(\vec{\mathbf{x}}) + \mathbf{b}^{k'}) = \vec{\mathbf{x}}$$

so when the $k$'th layer is trained the $k'$'th layer is simply the identity function for any $k' > k$. As a result, we have $\hat{\mathbf{f}}^k(\mathbf{x}) = W^D\Gamma^k(\mathbf{x})$.

Our first observation in the proof of Theorem A.1 is that the $k$'th step of algorithm 1 (i.e., obtaining $\hat{\mathbf{f}}^k$ from $\hat{\mathbf{f}}^{k-1}$) is essentially equivalent to learning a linear classifier on top of random features extension of that data representation $\vec{\mathbf{x}} \mapsto \hat{\mathbf{f}}^{k-1}(\vec{\mathbf{x}})$. Specifically, define an input space embedding $\Phi^{k-1} : \mathcal{X}^G \to \mathbb{R}^{q,G}$ by

$$\Phi^{k-1}(\vec{\mathbf{x}}) = \sigma(W_1^k E(\Gamma^{k-1}(\vec{\mathbf{x}})) + \mathbf{b}^k) = \sigma(W_1^k E((W^D)^{-1}\hat{\mathbf{f}}^k(\mathbf{x})) + \mathbf{b}^k) =$$

For $\mathbf{w} \in \mathbb{R}^q$ we define

$$\hat{\mathbf{f}}_{j,\mathbf{w}}^k(\vec{\mathbf{x}}) = \hat{\mathbf{f}}_j^{k-1}(\vec{\mathbf{x}}) + \mathbf{w}^\top\Phi^{k-1}(\vec{\mathbf{x}})$$

We have that

**Lemma A.2.** *For any* $D - 1 \geq k \geq 1$ $\hat{\mathbf{f}}_j^k = \hat{\mathbf{f}}_{j,\mathbf{w}}^k$ *where* $\mathbf{w}$ *is an* $\frac{\epsilon_{\text{opt}}}{2}$-*minimizer of the convex objective*

$$\ell_{S,j}^k(\mathbf{w}) = \ell_{S,j}\left(\hat{\mathbf{f}}_{j,\mathbf{w}}^k\right) + \frac{\epsilon_{\text{opt}}}{2}\|\mathbf{w}\|^2$$

*over* $\mathbf{w} \in \mathbb{R}^q$. *Furthermore,*

$$\ell_{S,j}(\hat{\mathbf{f}}^k) \leq \ell_{S,j}^k(\mathbf{w}^*) + \frac{\epsilon_{\text{opt}}}{2}\|\mathbf{w}^*\|^2 + \frac{\epsilon_{\text{opt}}}{2}$$

*Proof.* When the $k$'th layer is trained, since all deeper layers during this training phase are the identity function, the output of the network as a function of $W_2^k$ (the parameters that are trained in the $k$'th step) is

$$G(W_2^k, \vec{\mathbf{x}}) = W^D \left( \Gamma_{k-1}(\vec{\mathbf{x}}) + W_2^k \Phi^{k-1}(\vec{\mathbf{x}}) \right) = \hat{\mathbf{f}}^{k-1}(\vec{\mathbf{x}}) + W^D W_2^k \Phi^{k-1}(\vec{\mathbf{x}})$$

In particular, if we denote by $\hat{W}_2^k$ the value of $W_2^k$ after the $k$'th layer is trained, then we have $\hat{\mathbf{f}}_j^k = \hat{\mathbf{f}}_{j,\mathbf{w}}^k$ where $\mathbf{w}$ is the $j$'th row of the matrix $W = W^D \hat{W}_2^k$. It remains therefore to show that $\mathbf{w}$ minimizes $\ell_{S,j}^k$. To this end, we note that at the $k$'th step algorithm 1 finds an $\frac{\epsilon_{\mathrm{opt}}}{2}$-minimizer of

$$L(W_2^k) = \frac{\epsilon_{\mathrm{opt}}}{2} \|W_2^k\|^2 + \frac{1}{m} \sum_{t=1}^m \sum_{j=1}^n \ell(\hat{\mathbf{f}}^{k-1}(\vec{\mathbf{x}}) + W^D W_2^k \Phi^{k-1}(\vec{\mathbf{x}}), \mathbf{y}_j^t)$$

As a result, $\hat{W} := W^D \hat{W}_2^k$ is an $\frac{\epsilon_{\mathrm{opt}}}{2}$-minimizer of

$$
\begin{aligned}
L'(W) = L((W^D)^{-1} W) \quad &= \quad \frac{\epsilon_{\mathrm{opt}}}{2} \|(W^D)^{-1} W\|^2 + \frac{1}{m} \sum_{t=1}^m \sum_{j=1}^n \ell(\hat{\mathbf{f}}_j^{k-1}(\vec{\mathbf{x}}) + W^d (W^D)^{-1} W \Phi^{k-1}(\vec{\mathbf{x}}), \mathbf{y}_j^t) \\
&\overset{W^D \text{ is orthogonal}}{=} \quad \frac{\epsilon_{\mathrm{opt}}}{2} \|W\|^2 + \frac{1}{m} \sum_{t=1}^m \sum_{j=1}^n \ell(\hat{\mathbf{f}}_j^{k-1}(\vec{\mathbf{x}}) + W \Phi^{k-1}(\vec{\mathbf{x}}), \mathbf{y}_j^t) \\
&= \quad \sum_{j=1}^n \left( \frac{\epsilon_{\mathrm{opt}}}{2} \|W_{j\cdot}\|^2 + \frac{1}{m} \sum_{t=1}^m \ell(\hat{\mathbf{f}}_j^{k-1}(\vec{\mathbf{x}}) + W_{j\cdot} \Phi^{k-1}(\vec{\mathbf{x}}), \mathbf{y}_j^t) \right) \\
&= \quad \sum_{j=1}^n \ell_{S,j}^k(W_{j\cdot})
\end{aligned}
$$

In particular, $\mathbf{w} = \hat{W}_{j\cdot}$ must be $\frac{\epsilon_{\mathrm{opt}}}{2}$-minimizer of $\ell_{S,j}^k$ Finally, since $\ell_{S,j}^k$ is $\epsilon_{\mathrm{opt}}$-strongly convex, Equation (2) implies that for any $\mathbf{w}^* \in \mathbb{R}^q$,

$$\ell_{S,j}(\hat{\mathbf{f}}^k) \leq \ell_{S,j}^k(\mathbf{w}^*) + \frac{\epsilon_{\mathrm{opt}}}{2} \|\mathbf{w}^*\|^2 + \frac{\epsilon_{\mathrm{opt}}}{2}$$

$\square$

With lemma A.2 at hand, we can present the strategy of the proof. Since the labels in $L_1$ are PTF of the input, we will learn them when the first layer is trained. That is, $\hat{\mathbf{f}}^1$ will predict the labels in $L_1$ correctly. The reason for that is that, roughly speaking, PTFs are efficiently learnable by training a linear classifier on top of random features embedding.

Since, $\hat{\mathbf{f}}^1$ predicts the labels in $L_1$ correctly, the labels in $L_2$ become a simple function of $\hat{\mathbf{f}}^1$. Concretely, PTF of $\mathrm{sign}(\hat{\mathbf{f}}^1)$. It is therefore tempting to try using the same reasoning as above in order to prove that after training the next layer, we will learn the labels in $L_2$, and more generally, that after $r$ layers are trained, the network will predict all labels correctly. This however won't work that smoothly: PTF of $\mathrm{sign}(\hat{\mathbf{f}}^1)$ is not necessarily learnable by training a linear classifier on top of random-features embedding on $\hat{\mathbf{f}}^1$. To circumvent this, we show that after the network predicts correctly a label $j$, the loss of this label keeps improving when training additional layers, so after training additional $O(B + 1/\xi)$ layers, the loss will be small enough to guarantee that the labels in $L_2$ are PTFs of $\hat{\mathbf{f}}^1$ (and not just of $\mathrm{sign}(\hat{\mathbf{f}}^1)$). Thus, after $O(B + 1/\xi)$ layers are trained, the network will predict the labels in $L_2$ correctly, and more generally, after $O(rB + r/\xi)$ layers are trained, the network will predict all the labels correctly.

The course of the proof will be as follows

1. We start with Lemma A.4 which shows that if a label $j$ is a large PTF of $\hat{\mathbf{f}}^k$ then $\hat{\mathbf{f}}^{k+1}$ will predict it correctly. To be more accurate, we show that if a robust version of $\ell_{S,j}(p \circ E \circ \hat{\mathbf{f}}^k)$ is small for a polynomial $p$, then $\ell_{S,j}(\hat{\mathbf{f}}^{k+1})$ is small.

2. We then continue with Lemma A.5 which uses Lemma A.4 to show that (i) $\ell_{S,j}(\hat{\mathbf{f}}^1)$ is small for any $j \in L_1$, (ii) for any $j \in [n]$, if $\ell_{S,j}(\hat{\mathbf{f}}^k)$ is small, then it will shrink exponentially as we train deeper layers and (iii) if $\ell_{S,j}(\hat{\mathbf{f}}^k)$ is very small for any $j \in L_{i-1}$, then $\ell_{S,j}(\hat{\mathbf{f}}^{k+1})$ is small for any $j \in L_i$.

3. Based Lemma A.5, we will prove Theorem A.1.

The carry out the first step, we will need some notation. First, we define the $\epsilon$ robust version of $\ell$ as

$$\ell^{\mathrm{rob},\epsilon}(z) = \max(\ell(z), \ell(z - \epsilon)) = \max_{0 \leq t \leq \epsilon} \ell(z - t) \tag{9}$$

Note that for $z \leq 1$ we have $\ell^{\mathrm{rob},\epsilon}(z) = \ell(z - \epsilon)$ while for $z < 0$ we have $\ell^{\mathrm{rob},\epsilon}(z) = \ell(z) = \infty$. Denote the Hermite expansion of $\sigma$ by

$$\sigma = \sum_{s=0}^{\infty} a_s h_s \tag{10}$$

Let $K'$ be the minimal integer $K' \geq K$ such that $a_{K'} \neq 0$ (such $K'$ exists as otherwise $\sigma$ is a polynomial, which contradicts the assumption that it is bounded and non-constant). For $\epsilon > 0$ define $\beta(\epsilon) = \beta_{\sigma,K',K}(\epsilon) < 1$ as the minimal positive number greater that $\frac{3}{4}$ such that if $\beta_{\sigma,K',K}(\epsilon) \leq \beta < 1$ then

$$\frac{\|\sigma\|}{a_{K'}} 2^{(K'+2)/2} \frac{1 - \beta^2}{\sqrt{1 - 2(1 - \beta^2)^2}} \leq \frac{\epsilon}{2}$$

Note that $\beta(\epsilon)$ is well defined as $h(\beta) := \frac{1 - \beta^2}{\sqrt{1 - 2(1 - \beta^2)^2}}$ is continuous near $\beta = 1$ and equals to $0$ at $\beta = 1$. In fact, since $h$ is differentiable near $\beta = 1$ we have that $1 - \beta(\epsilon) = \Omega\left(\epsilon 2^{-K'} \frac{a_{K'}}{\|\sigma\|}\right)$. In particular, for fixed $\sigma, K', K$ we have that $1 - \beta(\epsilon) = \Omega(\epsilon)$. Define also

$$\delta(\epsilon, \beta, q, M, n) = \delta_{\sigma,K',K}(\epsilon, \beta, q, M, n) = \begin{cases} 1 & \frac{4\|\sigma\|_\infty}{\epsilon\sqrt{q}} \cdot \frac{1}{a_{K'}^2 \beta^{2K'-2K}} \left(\frac{n}{1-\beta^2}\right)^K M^2 > 1 \\ 2\exp\left(-q \cdot \frac{a_{K'}^4 \beta^{4K'-4K}(1-\beta^2)^{2K}\epsilon^4}{512n^{2K}M^4\|\sigma\|_\infty^4}\right) & \text{otherwise} \end{cases}$$

Note that for fixed $\sigma, K', K$ and $1 - \beta = \Omega(\epsilon)$ we have

$$\delta(\epsilon, \beta, q, M, n) = \exp\left(-\Omega\left(q \cdot \frac{\epsilon^{2K+4}}{n^{2K}M^4}\right)\right) \tag{11}$$

We will need the following Lemma that is proved at the end of section D, and shows that it is possible to approximate a polynomial by composing a random layer, and a linear function.

**Lemma A.3.** *Fix $\mathcal{X} \subset [-1,1]^n$, a degree $K$ polynomial $p : \mathcal{X} \to [-1,1]$, $K' \geq K$ and $\epsilon > 0$. Let $(W, \mathbf{b}) \in \mathbb{R}^{q \times n} \times \mathbb{R}^q$ be $\beta$-Xavier pair for $1 > \beta \geq \beta_{\sigma,K',K}(\epsilon)$. Then there is a vector $\mathbf{w} = \mathbf{w}(W, \mathbf{b}) \in \mathbb{B}^q$ such that*

$$\forall \mathbf{x} \in \mathcal{X}, \ \Pr\left(|\langle \mathbf{w}, \sigma(W\mathbf{x} + \mathbf{b})\rangle - p(\mathbf{x})| \geq \epsilon\right) \leq \delta_{\sigma,K',K}(\epsilon, \beta, q, \|p\|_{\mathrm{co}}, n)$$

We are now ready to show that if there a polynomial $p : \mathbb{R}^{wn} \to \mathbb{R}$ such that $\ell_{S,j}^{\mathrm{rob},\epsilon_1}(p \circ E_g \circ \hat{\mathbf{f}}^k)$ is small, then w.h.p. $\ell_{S,j}(\hat{\mathbf{f}}^{k+1})$ will be small as well.

**Lemma A.4.** *Fix $\epsilon_1 > 0$, $1 > \beta > \beta(\epsilon_1/2)$ and a polynomial $p : \mathbb{R}^{wn} \to \mathbb{R}$. Given that $\ell_{S,j}^{\mathrm{rob},\epsilon_1}(p \circ E_g \circ \hat{\mathbf{f}}^k) \leq \epsilon$, we have that $\ell_{S,j}(\hat{\mathbf{f}}^{k+1}) \leq \epsilon + \epsilon_{\mathrm{opt}}$ w.p. $1 - m|G|\delta(\epsilon_1/2, \beta, q, \|p\|_{\mathrm{co}} + 1, wn)$*

*Proof.* By lemma A.2 we have $\ell_{S,j}(\hat{\mathbf{f}}^{k+1}) \leq \ell_{S,j}\left(\hat{\mathbf{f}}_{j,\mathbf{w}^*}^{k+1}\right) + \frac{\epsilon_{\mathrm{opt}}}{2}\|\mathbf{w}^*\|^2 + \frac{\epsilon_{\mathrm{opt}}}{2}$ for any $\mathbf{w}^* \in \mathbb{R}^q$. Thus, it is enough to show that w.p. $1 - m|G|\delta(\epsilon_1/2, \beta, q, \|p\|_{\mathrm{co}} + 1, wn) =: 1 - \delta$ over the choice of $W_1^k$ there is $\mathbf{w}^* \in \mathbb{B}^d$ such that $\ell_{S,j}\left(\hat{\mathbf{f}}_{j,\mathbf{w}^*}^{k+1}\right) \leq \epsilon$. By the definition of $\ell_{S,j}^{\mathrm{rob},\epsilon_1}$ it is enough to show that w.p. $1 - \delta$ there is $\mathbf{w}^* \in \mathbb{B}^d$ such that

$$y_{j,g}^t \cdot p \circ E_g \circ \hat{\mathbf{f}}^k(\vec{\mathbf{x}}^t) - \epsilon_1 \leq y_{j,g}^t \cdot \hat{f}_{j,g,\mathbf{w}^*}^{k+1}(\vec{\mathbf{x}}^t) \leq y_{j,g}^t \cdot p \circ E_g \circ \hat{\mathbf{f}}^k(\vec{\mathbf{x}}^t) \tag{12}$$

for any $t$ and $g$. Since $y_{j,g}^t \cdot p \circ E_g \circ \hat{\mathbf{f}}^k(\vec{\mathbf{x}}^t) \geq \epsilon_1$ (as otherwise we will have $\ell_{S,j}^{\mathrm{rob},\epsilon_1}(p \circ E_g \circ \hat{\mathbf{f}}^k) = \infty$), it is enough to show that w.p. $1 - \delta$ there is $\tilde{\mathbf{w}}^* \in \mathbb{B}^d$ such that

$$\left| p \circ E_g \circ \hat{\mathbf{f}}^k(\vec{\mathbf{x}}^t) - \hat{f}_{j,g,\tilde{\mathbf{w}}^*}^{k+1}(\vec{\mathbf{x}}^t) \right| \leq \frac{\epsilon_1}{2}$$

for any $t$ and $g$. Indeed, in this case Equation (12) holds true for $\mathbf{w}^* = \frac{\tilde{\mathbf{w}}^*}{1+\epsilon_1/2}$. Finally, since

$$\hat{f}^{k+1}_{j,g,\mathbf{w}^*}(\vec{\mathbf{x}}) = \hat{f}^k_{j,g}(\vec{\mathbf{x}}) + \left\langle \mathbf{w}^*, \sigma(W^{k+1}_1 E_g \circ \hat{\mathbf{f}}^k(\vec{\mathbf{x}}) + \mathbf{b}^{k+1}) \right\rangle$$

it is enough to show that w.p. $1 - \delta$ there is $\tilde{\mathbf{w}}^* \in \mathbb{B}^d$ such that

$$\left| \tilde{p} \circ E_g \circ \hat{\mathbf{f}}^k(\vec{\mathbf{x}}^t) - \left\langle \mathbf{w}^*, \sigma(W^{k+1}_1 E_g \circ \hat{\mathbf{f}}^k(\vec{\mathbf{x}}^t) + \mathbf{b}^{k+1}) \right\rangle \right| \leq \frac{\epsilon_1}{2}$$

for the polynomial $\tilde{p}(\mathbf{x}^1 | \ldots | \mathbf{x}^w) = p(\mathbf{x}^1 | \ldots | \mathbf{x}^w) - x^1_j$ (note that $\tilde{p}(E(\hat{\mathbf{f}}^k(\vec{\mathbf{x}}))) = p(E(\hat{\mathbf{f}}^k(\vec{\mathbf{x}}))) - \hat{f}^k_j(\vec{\mathbf{x}})$ and that $\|\tilde{p}\|_{\mathrm{co}} \leq \|p\|_{\mathrm{co}} + 1$), and for any $t$ and $g$. The existence of such $\mathbf{w}^*$ w.p. $1 - \delta$ follows from Lemma A.3 and a union bound over $X = \{E_g \circ \hat{\mathbf{f}}^k(\vec{\mathbf{x}}^t) : g \in G, t \in [m]\}$ $\qquad\square$

We continue with the following Lemma which quantitatively describes how the loss of the different labels improves when training deeper and deeper layers.

**Lemma A.5.** *Let* $\gamma = \frac{1}{32} \min\left(\frac{1}{B}, \xi\right)$ *Assume that* $1 > \beta \geq \beta(\gamma/2)$ *and let* $\delta = m|G|\delta(\gamma/2, \beta, q, \|p\|_{\mathrm{co}} + 5, wn)$ *Then,*

- *For any* $j \in L_1$, *w.p.* $1 - \delta$, $\ell_{S,j}(\mathbf{f}^1) \leq \frac{1}{4m|G|} + \epsilon_{\mathrm{opt}}$

- *Given that* $\ell_{S,j}(\mathbf{f}^k) \leq \frac{1}{2m|G|}$ *we have that* $\ell_{S,j}(\mathbf{f}^{k+1}) \leq e^{-\gamma}\ell_{S,j}(\mathbf{f}^k) + \epsilon_{\mathrm{opt}}$ *w.p.* $1 - \delta$. *Furthermore, if* $\epsilon_{\mathrm{opt}} \leq \frac{1-e^{-\gamma}}{2m|G|}$ *then w.p.* $1 - t\delta$ *we have* $\ell_{S,j}(\mathbf{f}^{k+t}) \leq e^{-\gamma t}\ell_{S,j}(\mathbf{f}^k) + \frac{1-e^{-\gamma t}}{1-e^{-\gamma}}\epsilon_{\mathrm{opt}}$.

- *Given that* $\ell_{S,j'}(\mathbf{f}^k) \leq \frac{\xi}{8m^2|G|^2}$ *for any* $j' \in L_{i-1}$ *we have that* $\ell_{S,j}(\mathbf{f}^{k+1}) \leq \frac{1}{4m|G|} + \epsilon_{\mathrm{opt}}$ *for any* $j \in L_i$ *w.p.* $1 - |L_i|\delta$

Before proving Lemma A.5 implies, we show that it implies Theorem A.1.

*Proof.* (of Theorem A.1) Choose $\beta = \beta(\gamma/2)$ (more generally, $1 > \beta \geq \beta(\gamma/2)$ such that $1 - \beta = \Omega(\gamma)$). Denote $\delta = m|G|\delta(\gamma/2, \beta, q, M + 5, wn)$ and note that by Equation (11) we have

$$\delta = m|G| \exp\left(-\Omega\left(q \cdot \frac{\gamma^{2K+4}}{(wn)^{2K}(M+1)^4}\right)\right)$$

Since $\epsilon_{\mathrm{opt}} \leq \frac{(1-e^{-\gamma})\xi}{16m^2|G|^2}$, we have that if $\ell_{S,j}(\mathbf{f}^k) \leq \frac{1}{2m|G|}$ then w.p. $1 - t\delta$

$$\ell_{S,j}(\mathbf{f}^{k+t}) \leq e^{-\gamma t}\ell_{S,j}(\mathbf{f}^k) + \frac{1}{1-e^{-\gamma}}\epsilon_{\mathrm{opt}} \leq \frac{e^{-\gamma t}}{2m|G|} + \frac{\xi}{16m^2|G|^2}$$

Choosing $t_0 = \left\lceil \frac{\ln(8m|G|/\xi)}{\gamma} \right\rceil$ we get

$$\ell_{S,j}(\mathbf{f}^{k+t_0}) \leq \frac{\xi}{8m^2|G|^2}$$

w.p. $1 - t_0\delta$. Hence, it is not hard to verify by induction on $1 \leq i \leq r$ that for any $j \in L_i$, if $k \geq i(t_0 + 1)$ then

$$\ell_{S,j}(\mathbf{f}^k) \leq \frac{\xi}{8m^2|G|^2}$$

w.p. $1 - nk\delta$ $\qquad\square$

To prove lemma A.5 we will use the following fact which is an immediate consequence of the definition of the loss.

*Fact* A.6.    • If $\ell_{S,j}(\hat{\mathbf{f}}) \leq \frac{\epsilon}{m|G|}$ then for any $t \in [m]$ and $g \in G$ we have $1 \geq \hat{f}_{j,g}(\vec{\mathbf{x}}^t) \cdot f^*_{j,g}(\vec{\mathbf{x}}^t) \geq \frac{(1-\epsilon)}{2B}$

- If $\ell_{S,j}(\hat{\mathbf{f}}) \leq \frac{\epsilon}{4m^2|G|^2}$ then for any $t \in [m]$ and $g \in G$ we have $1 \geq \hat{f}_{j,g}(\vec{\mathbf{x}}^t) \cdot f^*_{j,g}(\vec{\mathbf{x}}^t) \geq (1-\epsilon)(1-\xi/2)$

- If for any $t \in [m]$ and $g \in G$ we have $1 \geq \hat{f}_{j,g}(\vec{\mathbf{x}}^t) \cdot f^*_{j,g}(\vec{\mathbf{x}}^t) \geq \frac{1}{B}$ then $\ell^{\mathrm{rob},1/2B}_{S,j}(\hat{\mathbf{f}}) \leq \frac{1}{4m|G|}$

We next prove lemma A.5.

*Proof.* (of lemma A.5) Let $p_1, \ldots p_n$ be polynomials that witness that $(\mathcal{L}, e)$ is an $(r, K, M, B, \xi)$-hierarchy for $\mathbf{f}^*$. We start with the first item. By the definition of hierarchy, we have that for any $t \in [m]$ and $g \in G$, $B \geq p_j(E_p(\mathbf{f}^0(\vec{\mathbf{x}}^t)))f_{j,g}(\vec{\mathbf{x}}^t) \geq 1$. Fact A.6 implies that for $\tilde{p}_j = \frac{1}{B}p_j$ we have $\ell^{\mathrm{rob},\gamma}_{S,j}(\tilde{p}_j \circ \hat{\mathbf{f}}^0) \leq \ell^{\mathrm{rob},1/2B}_{S,j}(\tilde{p}_j \circ \hat{\mathbf{f}}^0) \leq \frac{1}{4m|G|}$. The first item therefore follows from Lemma A.4.

The third item is proved similarly. If $\ell_{S,j'}(\mathbf{f}^k) \leq \frac{\xi}{8m^2|G|^2}$ for any $j' \in L_{i-1}$ then Fact A.6 implies that for any $j' \in L_{i-1}$, $t \in [m]$ and $g \in G$ we have

$$1 \geq y^t_{j',g}\hat{f}^k_{j',g}(\vec{\mathbf{x}}^t) \geq (1 - \xi/2)(1 - \xi/2) \geq 1 - \xi$$

Hence, by the definition of hierarchy, we have that for any $t \in [m]$ and $g \in G$, $B \geq p_j(E_p(\hat{\mathbf{f}}^k(\vec{\mathbf{x}}^t)))f_{j,g}(\vec{\mathbf{x}}^t) \geq 1$. Fact A.6 now implies that for $\tilde{p}_j = \frac{1}{B}p_j$ we have $\ell^{\mathrm{rob},\gamma}_{S,j}(\tilde{p}_j \circ \hat{\mathbf{f}}^k) \leq \ell^{\mathrm{rob},1/2B}_{S,j}(\tilde{p}_j \circ \hat{\mathbf{f}}^k) \leq \frac{1}{4m|G|}$. The third item therefore follows from Lemma A.4.

It remains to prove the second item. Define $q : \mathbb{R}^n \to \mathbb{R}$ by $q(\mathbf{x}) = 1.5x_j - 0.5x_j^3$. By lemma A.4 it is enough to show that

$$\ell^{\mathrm{rob},\gamma}_{S,j}(q \circ \hat{\mathbf{f}}^k) \leq e^{-\gamma}\ell_{S,j}(\hat{\mathbf{f}}^k) \tag{13}$$

To do so, we note that since $\ell_{S,j}(\hat{\mathbf{f}}^k) \leq \frac{1}{2m|G|}$ then Fact A.6 implies that $\forall t, g,\ y^t_{j,g}\hat{f}^k_{j,g}(\vec{\mathbf{x}}^t) \geq 1/(4B)$. Now, since $q$ is odd we have

$$\ell\left(y^t_{j,g}q\left(\hat{f}^k_{j,g}(\vec{\mathbf{x}}^t)\right)\right) = \ell\left(q\left(y^t_{j,g} \cdot \hat{f}^k_{j,g}(\vec{\mathbf{x}}^t)\right)\right)$$

Equation (13) therefore follows from the following claim

**Claim 1.** Let $\tilde{q}(x) = 1.5x - 0.5x^3$. Then, for any $\frac{1}{4B} \leq x \leq 1$ we have $\ell^{\mathrm{rob},\gamma}(\tilde{q}(x)) = \ell(\tilde{q}(x) - \gamma) \leq e^{-\gamma}\ell(x)$.

*Proof.* Denote $x' = \min(x, 1 - \xi/2)$ and note that $\ell(x) = \ell(x')$ and that

$$\tilde{q}(x') - x' = \frac{1}{2}x'(1 - x'^2) = \frac{1}{2}x'(1 - x')(1 + x') \geq \frac{1}{2}x'(1 - x') \geq \frac{1}{4}\min\left(1/4B, 1/2\xi\right) \geq 2\gamma \tag{14}$$

Now, we have

$$
\begin{aligned}
\ell(\tilde{q}(x) - \gamma) \quad &\overset{\substack{x' \leq x \\ \text{Eq. (14)}}}{\leq} \quad \ell(\tilde{q}(x') - \gamma) \\
&\overset{}{\leq} \quad \ell(x' + \gamma) \\
&= \quad \ell\left(\frac{1 - x' - \gamma}{1 - x'}x' + \frac{\gamma}{1 - x'}\right) \\
&\overset{\text{Convexity and } \frac{\gamma}{1-x'} \leq 1}{\leq} \quad \frac{1 - x' - \gamma}{1 - x'}\ell(x') + \frac{\gamma}{1 - x'}\ell(1) \\
&\overset{\ell(1) = 0 \text{ and } \ell(x') = \ell(x)}{=} \quad \frac{1 - x' - \gamma}{1 - x'}\ell(x) \\
&\leq \quad e^{-\gamma}\ell(x)
\end{aligned}
$$

$\square$

$\square$

# B. More Preliminaries

In the sequel we denote by $(\mathbb{R}^n)^{\otimes t}$ the space of order $t$ real tensors whose all axes has dimension $n$. We equip it with the inner product $\langle A, B \rangle = \sum_{1 \leq i_1, \ldots, i_t \leq n} A_{i_1,\ldots,i_t}B_{i_1,\ldots,i_t}$. For $\mathbf{x} \in \mathbb{R}^d$ we denote by $\mathbf{x}^{\otimes t} \in (\mathbb{R}^n)^{\otimes t}$ the tensor whose $(i_1, \ldots, i_t)$ entry is $\prod_{j=1}^{t}x_{i_j}$. We note that $\langle \mathbf{x}^{\otimes t}, \mathbf{y}^{\otimes t} \rangle = \langle \mathbf{x}, \mathbf{y} \rangle^t$.

## B.1. Concentration of Measure

We will use the Chernoff and Hoeffding's inequalities:

**Lemma B.1** (Hoeffding). *Let $X_1, \ldots, X_q \in [-B, B]$ be i.i.d. with mean $\mu$. Then, for any $\epsilon > 0$ we have*

$$\Pr\left(\left|\frac{1}{q}\sum_{i=1}^{q}X_i - \mu\right| \geq \epsilon\right) \leq 2e^{-\frac{q\epsilon^2}{2B^2}}$$

**Lemma B.2** (Chernoff). *Let $X_1, \ldots, X_q \in \{0, 1\}$ be i.i.d. with mean $\mu$. Then, for any $0 \leq \epsilon \leq \mu$ we have*

$$\Pr\left(\left|\frac{1}{q}\sum_{i=1}^{q}X_i - \mu\right| \geq \epsilon\right) \leq 2e^{-\frac{q\epsilon^2}{3\mu}}$$

We will also need to following version of Chernoff's bound.

**Lemma B.3.** *Let $X_1, \ldots, X_q \in \{-1, 1, 0\}$ be i.i.d. random variables with mean $\mu$. Then for $\epsilon \leq \frac{\min(\Pr(X_i=1),\Pr(X_i=-1))}{2|\mu|}$,*
$\Pr\left(\left|\frac{1}{q|\mu|}\sum_{i=1}^{n}X_i - \frac{\mu}{|\mu|}\right| \geq \epsilon\right) \leq 4e^{-\frac{q\epsilon^2|\mu|^2}{12\Pr(X_i\neq 0)}}$

*Proof.* (of Lemma B.3) Let $X_i^+ = \max(X_i, 0)$ and $\mu_+ = \mathbb{E}X_i^+ = \Pr(X_i = 1)$. Similarly, let $X_i^- = \max(-X_i, 0)$ and $\mu_- = \mathbb{E}X_i^- = \Pr(X_i = -1)$. By Chernoff bound (Lemma B.2) we have for $0 \leq \delta \leq 1$

$$\Pr\left(\left|\frac{1}{q}\sum_{i=1}^{n}X_i^+ - \mu_+\right| \geq \delta\mu_+\right) \leq 2e^{-\frac{q\delta^2\mu_+}{3}}$$

Hence,

$$\Pr\left(\left|\frac{1}{q|\mu|}\sum_{i=1}^{n}X_i^+ - \frac{\mu_+}{|\mu|}\right| \geq \delta\frac{\mu_+}{|\mu|}\right) \leq 2e^{-\frac{q\delta^2\mu_+}{3}}$$

Defining $\epsilon = \delta\frac{\mu_+}{|\mu|}$ we get for $\epsilon \leq \frac{\mu_+}{|\mu|}$

$$\Pr\left(\left|\frac{1}{q|\mu|}\sum_{i=1}^{n}X_i^+ - \frac{\mu_+}{|\mu|}\right| \geq \epsilon\right) \leq 2e^{-\frac{q\epsilon^2|\mu|^2}{3\mu_+}} \leq 2e^{-\frac{q\epsilon^2|\mu|^2}{3\Pr(X_i\neq 0)}}$$

A similar argument implies that for $\epsilon \leq \frac{\mu_-}{|\mu|}$ we have

$$\Pr\left(\left|\frac{1}{q|\mu|}\sum_{i=1}^{n}X_i^- - \frac{\mu_-}{|\mu|}\right| \geq \epsilon\right) \leq 2e^{-\frac{q\epsilon^2|\mu|^2}{3\Pr(X_i\neq 0)}}$$

As a result for $\epsilon \leq \frac{\min(\mu_+,\mu_-)}{2|\mu|}$ we have

$$\Pr\left(\left|\frac{1}{q|\mu|}\sum_{i=1}^{n}X_i - \frac{\mu}{|\mu|}\right| \geq \epsilon\right) \leq \Pr\left(\left|\frac{1}{q|\mu|}\sum_{i=1}^{n}X_i^+ - \frac{\mu_+}{|\mu|}\right| \geq \frac{\epsilon}{2}\right) + \Pr\left(\left|\frac{1}{q|\mu|}\sum_{i=1}^{n}X_i^- - \frac{\mu_-}{|\mu|}\right| \geq \frac{\epsilon}{2}\right)$$

$$\leq 4e^{-\frac{q\epsilon^2|\mu|^2}{12\Pr(X_i\neq 0)}}$$

$\square$

## B.2. Misc Lemmas

We will use the following asymptotics of binomials Coefficients, which follows from Stirling's approximation

**Lemma B.4.** *We have $\frac{\binom{2k}{k}}{2^{2k}} \sim \frac{1}{\sqrt{\pi k}}$*

We will also need the following approximation of the sign function using polynomials.

**Lemma B.5.** *Let $0 < \xi < 1$ and $\epsilon > 0$. There is a polynomial $p : \mathbb{R} \to \mathbb{R}$ such that*

- $p([-1, 1]) \subseteq [-1, 1]$

- *For any $x \in [-1, 1] \setminus [-\xi, \xi]$ we have $|p(\mathbf{x}) - \mathrm{sign}(\mathbf{x})| \leq \epsilon$.*

- $\deg(p) = O\left(\frac{\log(1/\epsilon)}{\xi}\right)$

- *$p$'s coefficients are all bounded by $2^{O\left(\frac{\log(1/\epsilon)}{\xi}\right)}$*

The existence of a polynomial that satisfies the first three properties is shown in (Diakonikolas et al., 2010). The bound on the coefficients (the last item) follows from Lemma 2.8. in (Sherstov, 2018) (see also here). Finally, we will use the following bound on the coefficient norm of a composition of a polynomial with a linear function.

**Lemma B.6.** *Fix a degree $K$ polynomial $p : \mathbb{R}^n \to \mathbb{R}$ and $A \in M_{n,m}$ whose rows has Euclidean norm at most $R$. Define $q(\mathbf{x}) = p(A\mathbf{x})$. Then, $\|q\|_{\mathrm{co}} \leq \|p\|_{\mathrm{co}} R^K (n+1)^{K/2}$*

*Proof.* Let $\mathbf{a}_i$ be the $i$'th row of $A$. Denote $p(\mathbf{x}) = \sum_{\alpha \in \{0,...,K\}^n, \|\alpha\|_1 \leq K} b_\alpha \mathbf{x}^\alpha$ and $e_\alpha(\mathbf{x}) = \prod_{i=1}^n \langle \mathbf{a}_i, \mathbf{x} \rangle^{\sigma_i}$. We have $q = \sum_{\alpha \in \{0,...,K\}^n, \|\alpha\|_1 \leq K} b_\alpha e_\alpha$. Hence,

$$\|q\|_{\mathrm{co}} \leq \sum_{\alpha \in \{0,...,K\}^n, \|\alpha\|_1 \leq K} |b_\alpha| \cdot \|e_\alpha\| \overset{\text{C.S.}}{\leq} \|p\|_{\mathrm{co}} \cdot \sqrt{\sum_{\alpha \in \{0,...,K\}^n, \|\alpha\|_1 \leq K} \|e_\alpha\|^2}$$

Finally

$$\|e_\alpha\|^2 = \left\|\mathbf{a}_1^{\otimes \sigma_1} \otimes \ldots \otimes \mathbf{a}_n^{\otimes \sigma_n}\right\|^2 = \prod_{i=1}^n \|\mathbf{a}_1\|^{2\sigma_i} \leq R^{2K}$$

$\square$

### B.3. A Generalization Result

It is well established that for "nicely behaved" function classes in which functions are defined by a vector of parameters, the sample complexity is proportional to the number of parameters. For instance, a function class of the form $\mathcal{F} = \{\mathbf{x} \mapsto F(\mathbf{w}, \mathbf{x}) : \mathbf{w} \in [-B, B]^p\}$ for a function $F$ that is $L$-Lipschitz in the first argument has realizable large margin sample complexity of $\tilde{O}\left(\frac{p}{\epsilon}\right)$. To be more precise, if there is a function in $\mathcal{F}$ with $\gamma$-error 0, then any algorithm that is guaranteed to return a function with empirical $\gamma$-error 0, enjoys this aforementioned sample complexity guarantee. We next slightly extend this fact, allowing $F$ to be random and allowing the algorithm to fail with some small probability.

**Lemma B.7.** *Suppose that $\mathcal{F} \subset (\mathbb{R}^n)^{\mathcal{X}}$ is a random function class such that*

- *There is a random function $F : [-B, B]^p \times \mathcal{X} \to \mathbb{R}^n$ such that $\mathcal{F} = \{\mathbf{x} \mapsto F(\mathbf{w}, \mathbf{x}) : \mathbf{w} \in [-B, B]^p\}$*

- *W.p. $1 - \delta_1$, for any $\mathbf{x} \in \mathcal{X}$, $\mathbf{w} \mapsto F(\mathbf{w}, \mathbf{x})$ is $L$-Lipschitz w.r.t. the $\ell^\infty$ norm.*

*Let $\mathcal{A}$ be an algorithm, and assume that for some $\mathbf{f}^* : \mathcal{X} \to \{\pm 1\}^n$, $\mathcal{A}$ has the property that on any $m$-points sample $S$ labeled by $\mathbf{f}^*$, it returns $\hat{\mathbf{f}} \in \mathcal{F}$ with $\mathrm{Err}_{S,\gamma}(\hat{\mathbf{f}}) = 0$ w.p. $1 - \delta_2$ (where the probability is over the randomness of $F$ and the internal randomness of $\mathcal{A}$). Then if $S$ is an i.i.d. sample labeled by $\mathbf{f}^*$ we have*

- $\mathrm{Err}_{\mathcal{D}}(\hat{\mathbf{f}}) \leq \epsilon$ *w.p.* $(LB/\gamma)^{O(p)}(1 - \epsilon)^m + \delta_1 + \delta_2$

- $\mathbb{E}_S \mathrm{Err}_{\mathcal{D}}(\hat{\mathbf{f}}) \leq O\left(\frac{p \ln(LB/\gamma) + \ln(m)}{m}\right) + \delta_1 + \delta_2$

*Proof.* (sketch) For $\hat{\mathbf{f}} : \mathcal{X} \to \mathbb{R}^n$ we define

$$\mathrm{Err}_{\mathcal{D},\gamma}(\hat{\mathbf{f}}) = \Pr_{\mathbf{x} \sim \mathcal{D}}\left(\exists i \in [n] \text{ s.t. } \hat{f}_i(\mathbf{x}) \cdot f_i(\mathbf{x}) < \gamma\right)$$

It is not hard to see that w.p. $1 - \delta_1$ there is $\tilde{\mathcal{F}} \subseteq \mathcal{F}$ of size $N = (LB/\gamma)^{O(p)}$ such that for any $\mathbf{g} \in \mathcal{F}$ there is $\tilde{\mathbf{g}} \in \tilde{\mathcal{F}}$ such that

$$\forall \mathbf{x} \in \mathcal{X}, \; \|\mathbf{g}(\mathbf{x}) - \tilde{\mathbf{g}}(\mathbf{x})\|_\infty \le \frac{\gamma}{2}$$

Let $A$ be the event that such $\tilde{\mathcal{F}}$ exists, that $\mathcal{A}$ return a function in $\mathcal{F}$ with $\mathrm{Err}_{S,\gamma}(\hat{\mathbf{f}}) = 0$, and that for any $\tilde{\mathbf{g}} \in \tilde{\mathcal{F}}$ with $\mathrm{Err}_{\mathcal{D},\gamma/2}(\tilde{\mathbf{g}}) \ge \epsilon$ we have $\mathrm{Err}_{S,\gamma/2}(\tilde{\mathbf{g}}) > 0$. We have that the probability of $A$ is at least $1 - \delta_1 - \delta_2 - N(1-\epsilon)^m$. Given $A$ we have for any $\mathbf{g} \in \mathcal{F}$,

$$\mathrm{Err}_{\mathcal{D}}(\mathbf{g}) \ge \epsilon \Rightarrow \mathrm{Err}_{\mathcal{D},\gamma/2}(\tilde{\mathbf{g}}) \ge \epsilon \Rightarrow \mathrm{Err}_{S,\gamma/2}(\tilde{\mathbf{g}}) > 0 \Rightarrow \mathrm{Err}_{S,\gamma}(\mathbf{g}) > 0$$

Thus, the probability that $\mathcal{A}$ return a function with error $\ge \epsilon$ is at most $N(1-\epsilon)^m + \delta_1 + \delta_2$ which proves the first part of the lemma. As for the second part, we note that we have

$$\mathbb{E}_S \mathrm{Err}_{\mathcal{D}}(\hat{\mathbf{f}}) \le \mathbb{E}_S[\mathrm{Err}_{\mathcal{D}}(\hat{\mathbf{f}})|A] + \Pr(A^{\complement}) \le \epsilon + N(1-\epsilon)^m + \delta_1 + \delta_2$$

Optimizing over $\epsilon$ we get $\mathbb{E}_S \mathrm{Err}_{\mathcal{D}}(\hat{\mathbf{f}}) \le \frac{\ln(Nm)}{m} + \delta_1 + \delta_2$ which proves the second part $\qquad\square$

## B.4. Kernels

The results we state next can be found in Chapter 2. of Schölkopf & Smola (2002). Let $\mathcal{X}$ be a set. A *kernel* is a function $k : \mathcal{X} \times \mathcal{X} \to \mathbb{R}$ such that for every $x_1, \dots, x_m \in \mathcal{X}$ the matrix $\{k(x_i, x_j)\}_{i,j}$ is positive semi-definite. A *kernel space* is a Hilbert space $\mathcal{H}$ of functions from $\mathcal{X}$ to $\mathbb{R}$ such that for every $x \in \mathcal{X}$ the linear functional $f \in \mathcal{H} \mapsto f(x)$ is bounded. The following theorem describes a one-to-one correspondence between kernels and kernel spaces.

**Theorem B.8.** *For every kernel $k$ there exists a unique kernel space $\mathcal{H}_k$ such that for every $x, x' \in \mathcal{X}$, $k(x, x') = \langle k(\cdot, x), k(\cdot, x') \rangle_{\mathcal{H}_k}$. Likewise, for every kernel space $\mathcal{H}$ there is a kernel $k$ for which $\mathcal{H} = \mathcal{H}_k$.*

We denote the norm and inner product in $\mathcal{H}_k$ by $\|\cdot\|_k$ and $\langle \cdot, \cdot \rangle_k$. The following theorem describes a tight connection between kernels and embeddings of $\mathcal{X}$ into Hilbert spaces.

**Theorem B.9.** *A function $k : \mathcal{X} \times \mathcal{X} \to \mathbb{R}$ is a kernel if and only if there exists a mapping $\Psi : \mathcal{X} \to \mathcal{H}$ to some Hilbert space for which $k(x, x') = \langle \Psi(x), \Psi(x') \rangle_{\mathcal{H}}$. In this case, $\mathcal{H}_k = \{f_{\Psi,\mathbf{v}} \mid \mathbf{v} \in \mathcal{H}\}$ where $f_{\Psi,\mathbf{v}}(x) = \langle \mathbf{v}, \Psi(x) \rangle_{\mathcal{H}}$. Furthermore, $\|f\|_k = \min\{\|\mathbf{v}\|_{\mathcal{H}} : f_{\Psi,\mathbf{v}}\}$ and the minimizer is unique.*

## B.5. Random Features Schemes

Let $\mathcal{X}$ be a measurable space and let $k : \mathcal{X} \times \mathcal{X} \to \mathbb{R}$ be a kernel. A *random features scheme* (RFS) for $k$ is a pair $(\psi, \mu)$ where $\mu$ is a probability measure on a measurable space $\Omega$, and $\psi : \Omega \times \mathcal{X} \to \mathbb{R}$ is a measurable function, such that

$$\forall \mathbf{x}, \mathbf{x}' \in \mathcal{X}, \quad k(\mathbf{x}, \mathbf{x}') = \mathbb{E}_{\omega \sim \mu} \psi(\omega, \mathbf{x}) \psi(\omega, \mathbf{x}'). \tag{15}$$

We often refer to $\psi$ (rather than $(\psi, \mu)$) as the RFS. We define $\|\psi\|_\infty = \sup_{\mathbf{x}} \|\psi(\cdot, \mathbf{x})\|_\infty$, and say that $\psi$ is $C$-bounded if $\|\psi\|_\infty \le C$. The random *q-embedding* generated from $\psi$ is the random mapping

$$\Psi_{\boldsymbol{\omega}}(\mathbf{x}) := (\psi(\omega_1, \mathbf{x}), \dots, \psi(\omega_q, \mathbf{x})),$$

where $\omega_1, \dots, \omega_q \sim \mu$ are i.i.d. The random *q-kernel* corresponding to $\Psi_{\boldsymbol{\omega}}$ is $k_{\boldsymbol{\omega}}(\mathbf{x}, \mathbf{x}') = \frac{\langle \Psi_{\boldsymbol{\omega}}(\mathbf{x}), \Psi_{\boldsymbol{\omega}}(\mathbf{x}') \rangle}{q}$. Likewise, the random *q-kernel space* corresponding to $\frac{1}{\sqrt{q}} \Psi_{\boldsymbol{\omega}}$ is $\mathcal{H}_{k_{\boldsymbol{\omega}}}$. We next discuss approximation of functions in $\mathcal{H}_k$ by functions in $\mathcal{H}_{k_{\boldsymbol{\omega}}}$. It would be useful to consider the embedding

$$\mathbf{x} \mapsto \Psi^{\mathbf{x}} \quad \text{where} \quad \Psi^{\mathbf{x}} := \psi(\cdot, \mathbf{x}) \in L^2(\Omega). \tag{16}$$

From (15) it holds that for any $\mathbf{x}, \mathbf{x}' \in \mathcal{X}, k(\mathbf{x}, \mathbf{x}') = \left\langle \Psi^{\mathbf{x}}, \Psi^{\mathbf{x}'} \right\rangle_{L^2(\Omega)}$. In particular, from Theorem B.9, for every $f \in \mathcal{H}_k$ there is a unique function $\check{f} \in L^2(\Omega)$ such that

$$\|\check{f}\|_{L^2(\Omega)} = \|f\|_k \tag{17}$$

and for every $\mathbf{x} \in \mathcal{X}$,

$$f(\mathbf{x}) = \left\langle \check{f}, \Psi^{\mathbf{x}} \right\rangle_{L^2(\Omega)} = \mathbb{E}_{\omega \sim \mu} \check{f}(\omega) \psi(\omega, \mathbf{x}) . \tag{18}$$

Let us denote $f_{\boldsymbol{\omega}}(\mathbf{x}) = \frac{1}{q} \sum_{i=1}^{q} \left\langle \check{f}(\omega_i), \psi(\omega_i, \mathbf{x}) \right\rangle$. From (18) we have that $\mathbb{E}_{\boldsymbol{\omega}}[f_{\boldsymbol{\omega}}(\mathbf{x})] = f(\mathbf{x})$. Furthermore, for every $\mathbf{x}$, the variance of $f_{\boldsymbol{\omega}}(\mathbf{x})$ is at most

$$\begin{aligned}
\frac{1}{q} \mathbb{E}_{\omega \sim \mu} \left| \check{f}(\omega) \psi(\omega, \mathbf{x}) \right|^2 &\leq \frac{\|\psi\|_\infty^2}{q} \mathbb{E}_{\omega \sim \mu} \left| \check{f}(\omega) \right|^2 \\
&= \frac{\|\psi\|_\infty^2 \|f\|_k^2}{q} .
\end{aligned}$$

An immediate consequence is the following corollary.

**Corollary B.10** (Function Approximation). *For all $\mathbf{x} \in \mathcal{X}$, $\mathbb{E}_{\boldsymbol{\omega}} |f(\mathbf{x}) - f_{\boldsymbol{\omega}}(\mathbf{x})|^2 \leq \frac{\|\psi\|_\infty^2 \|f\|_k^2}{q}$.*

Now, if $\mathcal{D}$ is a distribution on $\mathcal{X}$ we get that

$$\mathbb{E}_{\boldsymbol{\omega}} \|f - f_{\boldsymbol{\omega}}\|_{2,\mathcal{D}} \overset{\text{Jensen}}{\leq} \sqrt{\mathbb{E}_{\boldsymbol{\omega}} \|f - f_{\boldsymbol{\omega}}\|_{2,\mathcal{D}}^2} = \sqrt{\mathbb{E}_{\boldsymbol{\omega}} \mathbb{E}_{\mathbf{x} \sim \mathcal{D}} |f(\mathbf{x}) - f_{\boldsymbol{\omega}}(\mathbf{x})|^2} = \sqrt{\mathbb{E}_{\mathbf{x}} \mathbb{E}_{\boldsymbol{\omega}} |f(\mathbf{x}) - f_{\boldsymbol{\omega}}(\mathbf{x})|^2} \leq \frac{\|\psi\|_\infty \|f\|_k}{\sqrt{q}}$$

Thus, $O\left(\frac{\|f\|_k^2}{\epsilon^2}\right)$ random features suffices to guarantee that $\mathbb{E}_{\boldsymbol{\omega}} \|f - f_{\boldsymbol{\omega}}\|_{2,\mathcal{D}} \leq \epsilon$. In this paper such an $\ell^2$ guarantee will not suffice, and we will need an approximation of functions in $\mathcal{H}_k$ by functions in $\mathcal{H}_{k_{\boldsymbol{\omega}}}$ w.r.t. the stronger $\ell^\infty$ norm. We next show this can be obtained, unfortunately with a quadratic growth in the required number of features. For $z \in \mathbb{R}$ we define $\langle z \rangle_B = \begin{cases} z & |z| \leq B \\ 0 & \text{otherwise} \end{cases}$. We will consider the following a truncated version of $f_{\boldsymbol{\omega}}$

$$f_{\boldsymbol{\omega},B}(\mathbf{x}) = \frac{1}{q} \sum_{i=1}^{q} \left\langle \check{f}(\omega_i) \right\rangle_B \cdot \psi(\omega_i, \mathbf{x})$$

Now, if $\psi$ is $C$-bounded we have that $f_{\boldsymbol{\omega},B}(\mathbf{x})$ is and average of $q$ i.i.d. $CB$-bounded random variables. By Hoeffding's inequality, we have

$$\Pr\left(|f_{\boldsymbol{\omega},B}(\mathbf{x}) - \mathbb{E}_{\boldsymbol{\omega}'} f_{\boldsymbol{\omega}',B}(x)| > \epsilon/2\right) \leq 2e^{-\frac{q\epsilon^2}{8B^2C^2}} \tag{19}$$

Likewise, we have

$$\begin{aligned}
|f(x) - \mathbb{E}_{\boldsymbol{\omega}'} f_{\boldsymbol{\omega}',B}(x)| &= |\mathbb{E}\left(f_{\boldsymbol{\omega}}(x) - f_{\boldsymbol{\omega},B}(x)\right)| \\
&= \left| \mathbb{E}\left(\check{f}(\omega) - \left\langle \check{f}(\omega) \right\rangle_B\right) \cdot \psi(\omega, \mathbf{x}) \right| \\
&= \left| \mathbb{E} \mathbb{1}_{|\check{f}(\omega)| > B} \check{f}(\omega) \psi(\omega, \mathbf{x}) \right| \\
&\leq \sqrt{\Pr(|\check{f}(\omega)| > B) \mathbb{E}\left(\check{f}(\omega)\psi(\omega,\mathbf{x})\right)^2} \\
&\leq \|\psi\|_\infty \sqrt{\Pr(|\check{f}(\omega)| > B) \mathbb{E}\left(\check{f}(\omega)\right)^2} \\
&= \frac{\|\psi\|_\infty \|f\|_k^2}{B}
\end{aligned}$$

We get that

**Lemma B.11.** *Let $f \in \mathcal{H}_k$ with $\|f\|_k \leq M$ and assume that, $\|\psi\|_\infty \leq C$. For $B = \frac{2CM^2}{\epsilon}$ we have*

$$\Pr\left(|f_{\boldsymbol{\omega},B}(\mathbf{x}) - f(\mathbf{x})| > \epsilon\right) \leq 2e^{-\frac{q\epsilon^4}{32M^4C^4}}$$

*Furthermore, the norm of weight vector vector defining $f_{\boldsymbol{\omega},B}$, i.e. $\mathbf{w} = \frac{1}{q}\left(\left\langle \check{f}(\omega_1)\right\rangle_B, \dots, \left\langle \check{f}(\omega_q)\right\rangle_B\right)$, satisfies*

$$\|\mathbf{w}\| \leq \frac{2CM^2}{\epsilon\sqrt{q}}$$

# C. Examples of Hierarchies and Proof Theorem 3.4

Fix $\mathcal{X} \subset [-1, 1]^n$, a proximity mapping $\mathbf{e} : G \to G^w$, and a collection of sets $\mathcal{L} = \{L_1, \ldots, L_r\}$ such that $L_1 \subseteq L_2 \subseteq \ldots \subseteq L_r = [n]$. So far, we have seen one formal example to a hierarchy: In the non-ensemble setting (i.e. $w = |G| = 1$) Example 3.2 shows that if any label depends on $K$ simpler labels, and the labels in the first level are $(K, 1)$-PTFs of the input, then $\mathcal{L}$ is an $(r, K, 1)$-hierarchy. In this section we expand our set of examples. We first show (Lemma C.1) that if $(\mathcal{L}, \mathbf{e})$ is an $(r, K, M)$-hierarchy then it is an $(r, K, 2M, B, \xi)$-hierarchy for suitable $B$ and $\xi$. Then, in section C.1, consider in more detail the case that each label depends on a few simpler labels, in a few locations, and show that the parameters obtained from Lemma C.1 can be improved in this case. Finally, in section C.2 we prove Theorem 3.4, showing that if all the labels are "random snippets" from a given circuit, and there is enough of them, then the target function has a low-complexity hierarchy.

**Lemma C.1.** *Any $(r, K, M)$-hierarchy of $\mathbf{f}^* : \mathcal{X}^G \to \{\pm 1\}^{n,G}$ is also an $(r, K, 2M, B, \xi)$-hierarchy for $\xi = \frac{1}{2(wn+1)^{\frac{K+1}{2}} KM}$ and $B = 2(w \max(n, d) + 1)^{K/2} M$*

Lemma C.1 follows immediately from the definition of hierarchy and the following lemma

**Lemma C.2.** *Any $(K, M)$-PTF $f : \mathcal{X} \to \{\pm 1\}$ is a $(K, 2M, B, \xi)$-PTF w.r.t. for $\xi = \frac{1}{2(n+1)^{\frac{K+1}{2}} KM}$ and $B = 2(n+1)^{k/2} M$*

Lemma C.2 is implied by Lemmas C.3 and C.4

**Lemma C.3.** *Let $p : \mathbb{R}^n \to \mathbb{R}$ be a degree $K$ polynomial. Then $p$ is $((n+1)^{\frac{K+1}{2}} K \|p\|_{\text{co}})$-Lipschitz in $[-1, 1]^n$ w.r.t. the $\| \cdot \|_\infty$ norm and satisfies $|p(\mathbf{x})| \leq (n+1)^{k/2} \|p\|_{\text{co}}$ for any $\mathbf{x} \in [-1, 1]^n$.*

*Proof.* Denote $p(\mathbf{x}) = \sum_{\alpha \in \{0, \ldots, K\}^n, \|\alpha\|_1 \leq K} a_\alpha \mathbf{x}^\alpha$. We have

$$\frac{\partial p}{\partial x_i} (\mathbf{x}) = \sum_{\alpha \in \{0, \ldots, K-1\}^n, \|\alpha\|_1 \leq K-1} a_{\alpha + \mathbf{e}_i} \cdot (\alpha_i + 1) \cdot \mathbf{x}^\alpha$$

This implies that for any $\mathbf{x} \in [-1, 1]^n$ we have

$$
\begin{aligned}
\left| \frac{\partial p}{\partial x_i} (\mathbf{x}) \right| &\leq \sum_{\alpha \in \{0, \ldots, K-1\}^n, \|\alpha\|_1 \leq K-1} |a_{\alpha + \mathbf{e}_i} \cdot (\alpha_i + 1) \cdot \mathbf{x}^\alpha| \\
&\leq K \sum_{\alpha \in \{0, \ldots, K-1\}^n, \|\alpha\|_1 \leq K-1} |a_{\alpha + \mathbf{e}_i}| \\
&\leq K \sqrt{(n+1)^{K-1}} \|p\|_{\text{co}}
\end{aligned}
$$

Hence, $\|\nabla p(\mathbf{x})\|_1 \leq nK \sqrt{(n+1)^{K-1}} \|p\|_{\text{co}} \leq K \sqrt{(n+1)^{K+1}} \|p\|_{\text{co}}$. Showing that $p$ is $((n+1)^{\frac{K+1}{2}} K \|p\|_{\text{co}})$-Lipschitz in $[-1, 1]^n$ w.r.t. the $\| \cdot \|_\infty$ norm. Likewise, for any $\mathbf{x} \in [-1, 1]^n$ we have

$$p(\mathbf{x}) \leq \sum_{\alpha \in \{0, \ldots, K\}^n, \|\alpha\|_1 \leq K} |a_\alpha| \leq 2(n+1)^{K/2} \|p\|_{\text{co}}$$

$\square$

**Lemma C.4.** *Assume that $f : \mathcal{X} \to \{\pm 1\}$ is $(K, M)$-PTF w.r.t. as witnessed by a polynomial $p : \mathbb{R}^n \to \mathbb{R}$ that is $L$-Lipschitz w.r.t. $\| \cdot \|_\infty$.*

- *If $p$ is bounded by $B$ is $\cup_{\mathbf{x} \in \mathcal{X}} \mathcal{B}_{1/(2L)}(\mathbf{x})$. Then, $f$ is $\left( K, 2M, 2B, \frac{1}{2L} \right)$-PTF witnessed by $2p$*

- *If $p$ is bounded by $B$ is $\mathcal{X}$. Then, $f$ is $\left( K, 2M, 2B + 1, \frac{1}{2L} \right)$-PTF witnessed by $2p$*

*Proof.* We first note the the second item follows form the first. Indeed, if $p$ is bounded by $B$ in $\mathcal{X}$ then $p$ is bounded by $B + 1/2$ is $\cup_{\mathbf{x} \in \mathcal{X}} \mathcal{B}_{1/(2L)}(\mathbf{x})$. To prove the first item we need to show that for any $\mathbf{x} \in \mathcal{X}$ and $\tilde{\mathbf{x}} \in \mathcal{B}_\xi(\mathbf{x})$ we have

$$2B \geq 2p(\tilde{\mathbf{x}}) f(\mathbf{x}) \geq 1$$

The left inequality is clear. For the right inequality we assume that $f(\mathbf{x}) = 1$ (the other case is similar). Since $\|\mathbf{x} - \tilde{\mathbf{x}}\|_\infty \leq \frac{1}{2L}$ we have

$$
\begin{aligned}
p(\tilde{\mathbf{x}}) &\geq p(\mathbf{x}) - |p(\tilde{\mathbf{x}}) - p(\mathbf{x})| \\
&\geq p(\mathbf{x}) - L \cdot \|\mathbf{x} - \tilde{\mathbf{x}}\|_\infty \\
&\geq 1 - \frac{L}{2L} \\
&= \frac{1}{2}
\end{aligned}
$$

$\square$

### C.1. Each Label Depends on $O(1)$ Simpler Labels

Assume now that $\mathcal{X} \subseteq \{\pm 1\}^d$, and that any label $j \in L_i$ depends on at most $K$ labels from $L_{i-1}$ in at most $K$ locations (of $K$ input locations if $i = 1$). That is, for any $j \in L_i$, there is a function $\tilde{f}_j : \{\pm 1\}^{wn} \to \{\pm 1\}$ (or $\tilde{f}_j : \{\pm 1\}^{dw} \to \{\pm 1\}$ if $i = 1$) that depends at most $K$ coordinates, from $\{kn + l : 0 \leq k \leq w - 1, \ l \in L_{i-1}\}$ (from $[dw]$ if $i = 1$), for which the following holds. For any $g \in G$, $f_{j,g}^*(\vec{\mathbf{x}}) = \tilde{f}_j(E_g(\mathbf{f}^*(\vec{\mathbf{x}})))$ (or $f_{j,g}^*(\vec{\mathbf{x}}) = f_j(E_g(\vec{\mathbf{x}}))$ if $i = 1$).

As in example 3.2, since any Boolean function depending on $K$ variables is a $(K, 1)$-PTF, we have that the functions $\tilde{f}_j$ are $(K, 1)$-PTFs, implying that $(\mathcal{L}, \mathbf{e})$ in an $(r, K, 1)$-hierarchy. Lemma C.1 implies that $(\mathcal{L}, \mathbf{e})$ is $(r, K, 2, B, \xi)$-hierarchy for $\xi = \frac{1}{2K(wn+1)^{(K+1)/2}}$ and $B = 2(w \max(n, d) + 1)^{K/2}$. The following lemma shows that this can be substantially improved.

**Lemma C.5.** *Any Boolean function depending on $K$ coordinates is a $(K, 2, 3, \xi)$-PTF for $\xi = \frac{1}{K2^{(K+2)/2}}$. As a result $(\mathcal{L}, \mathbf{e})$ is $(r, K, 2, 3, \xi)$-hierarchy.*

Lemma C.5 follows from the following Lemma together with Lemma C.4

**Lemma C.6.** *Let $f : \{\pm 1\}^K \to \{\pm 1\}$ and let $F(\mathbf{x}) = \sum_{A \subseteq [K]} a_A \mathbf{x}^A$ be its standard multilinear extension. Then, $F$ is $(K2^{K/2})$-Lipschitz in $[-1, 1]^K$ w.r.t. the $\|\cdot\|_\infty$ norm.*

*Proof.* For $\mathbf{x} \in [-1, 1]^K$ we have

$$
\left| \frac{\partial F}{\partial x_i} \right| = \left| \sum_{i \in A \subseteq [K]} a_A \mathbf{x}^A \right| \leq \sum_{i \in A \subseteq [K]} |a_A| \leq \sum_{i \in A \subseteq [K]} |a_A|
$$

Hence,

$$
\|\nabla F(\mathbf{x})\|_1 \leq \sum_{A \subseteq [K]} |A| \, |a_A| \overset{\text{Cauchy Schwartz}}{\leq} K2^{K/2}
$$

$\square$

The following Lemma shows that $\xi$ and $B$ can be improved even further, at the expense of the degree and the coefficient norm.

**Lemma C.7.** *For any $0 < \xi < 1 < B$, any Boolean function depending on $K$ coordinates is a $(K', M, B, \xi)$-PTF for or $K' = O\left(\frac{K^2 + K \log((B+1)/(B-1))}{1 - \xi}\right)$ and $M = 2^{O\left(\frac{K^2 + K \log((B+1)/(B-1))}{1-\xi}\right)}$. As a result $(\mathcal{L}, \mathbf{e})$ is $(r, K', M, B, \xi)$-hierarchy*

*Proof.* Fix $f : \{\pm 1\}^K \to \{\pm 1\}$. We need to show that $f$ is a $(K', M, B, \xi)$-PTF. Let $\epsilon = \frac{B-1}{B+1}$. By Lemma B.5 there is a uni-variate polynomial $q$ of degree $O\left(\frac{K + \log(1/\epsilon)}{1 - \xi}\right)$ such that $q([-1, 1]) \subseteq [-1, 1]$, for any $y \in [-1, 1] \setminus [-1 + \xi, 1 - \xi]$ we have $|q(y) - \text{sign}(y)| \leq \frac{\epsilon}{K2^{K/2}}$, and the coefficients of $q$ are all bounded by $2^{O\left(\frac{K + \log(1/\epsilon)}{1 - \xi}\right)}$. Consider now the polynomial $\tilde{p}(\mathbf{x}) = F(q(\mathbf{x}))$ where $F$ is the multilinear extension on $f$. It is not hard to verify that $\deg(\tilde{p}) \leq \deg(q)K =$

$O\left(\frac{K^2 + K \log(1/\epsilon)}{1-\xi}\right)$ and that $\|\tilde{p}\|_{\text{co}} \leq 2^{O\left(\frac{K^2 + K \log(1/\epsilon)}{1-\xi}\right)}$. Finally, fix $\mathbf{x} \in \{\pm 1\}^K$ and $\tilde{\mathbf{x}} \in \mathcal{B}_\xi(\mathbf{x})$. Note that $\mathbf{x} = \text{sign}(\tilde{\mathbf{x}})$. Since $F$ is $K2^{k/2}$-Lipschitz w.r.t. the $\|\cdot\|_\infty$ norm in $[-1,1]^K$ (lemma C.6) we have

$$|\tilde{p}(\tilde{\mathbf{x}}) - f(\mathbf{x})| = |\tilde{p}(\tilde{\mathbf{x}}) - f(\text{sign}(\tilde{\mathbf{x}}))| = |F(q(\tilde{\mathbf{x}})) - F(\text{sign}(\tilde{\mathbf{x}}))| \leq \|q(\tilde{\mathbf{x}}) - \text{sign}(\tilde{\mathbf{x}})\|_\infty \leq \epsilon$$

Since $f(\mathbf{x}) \in \{\pm 1\}$ this implies that

$$1 + \epsilon \geq \tilde{p}(\tilde{\mathbf{x}})f(\mathbf{x}) \geq 1 - \epsilon$$

Taking $p(x) = \frac{1}{1-\epsilon}\tilde{p}(x)$ and noting that $B = \frac{1+\epsilon}{1-\epsilon}$ we get

$$B \geq p(\tilde{\mathbf{x}})f(\mathbf{x}) \geq 1$$

which implies that $f$ is a $(K', M, B, \xi)$-PTF. $\qquad\square$

## C.2. Proof of Theorem 3.4

In this section we will prove (a slightly extended version of) Theorem 3.4. We first recall and slightly extend the setting. Fix a domain $\mathcal{X} \subseteq \{\pm 1\}^d$ and a sequence of functions $G^i : \{\pm 1\}^d \to \{\pm 1\}^d$ for $1 \leq i \leq r$. We assume that $G^0(\mathbf{x}) = \mathbf{x}$, and for any depth $i \in [r]$ and coordinate $j \in [d]$, we have

$$\forall \mathbf{x} \in \mathcal{X}, \quad G^i_j(\mathbf{x}) = p^i_j(G^{i-1}(\mathbf{x})), \tag{20}$$

where $p^i_j : \{\pm 1\}^d \to \{\pm 1\}$ is a function whose multi-linear extension is a polynomial of degree at most $K$. Furthermore, we assume this extension is $L$-Lipschitz in $[-1,1]^d$ with respect to the $\ell_\infty$ norm (if $p^i_j$ depends on $K$ coordinates, as in the problem description in section 3.1, Lemma C.6 implies that this holds with $L = K2^{K/2}$). Fix an integer $q$. We assume that for every depth $i \in [r]$, there are $q$ auxiliary labels $f^*_{i,j}$ for $1 \leq j \leq q$, each of which is a signed Majority of an odd number of components of $G^i$. Moreover, we assume these functions are random. Specifically, prior to learning, the labeler independently samples $qr$ functions such that for any $i \in [r]$ and $j \in [q]$,

$$f^*_{i,j}(\mathbf{x}) = \text{sign}\left(\sum_{l=1}^d w^{i,j}_l G^i_l(\mathbf{x})\right), \tag{21}$$

where the weight vectors $\mathbf{w}^{i,j} \in \mathbb{R}^d$ are independent uniform vectors chosen from

$$\mathcal{W}_{d,k} := \left\{\mathbf{w} \in \{-1,0,1\}^d : \sum_{l=1}^d |w_l| = k\right\}$$

for some odd integer $k$. The following theorem, which slightly extends Theorem 3.4, shows that if $q \gg dL^2 \log(|\mathcal{X}|)$, then with high probability over the choice of $\mathbf{f}^*$, the target function $\mathbf{f}^*$ has an $(r, K, O(kd^K), 2k+1)$-hierarchy.

**Theorem C.8.** *W.p.* $1 - 4drq|\mathcal{X}|e^{-\Omega\left(\frac{q}{L^2 k^2 d}\right)}$ *the function* $\mathbf{f}^*$ *has* $(r, K, O(kd^K), 2k+1)$-*hierarchy*

In order to prove Theorem C.8 it is enough to show that for any $i \in [r]$ and $j \in [q]$, $f^*_{i,j}$ is a $(K, O(kd^K), 2k+1)$-PTF of

$$\Psi_{i-1}(\mathbf{x}) = (f^*_{i-1,1}(\mathbf{x}), \ldots, f^*_{i-1,q}(\mathbf{x}))$$

By equations (21) and (20) we have

$$f^*_{i,j}(\mathbf{x}) = \text{sign}\left(\sum_{l=1}^d w^{i,j}_l p^i_l(G^{i-1}(\mathbf{x}))\right)$$
$$=: \text{sign}\left(q(G^{i-1}(\mathbf{x}))\right)$$

Hence, $f^*_{i,j}$ is $(K, k)$-PTF of $G^{i-1}$, as witnessed by $q$ (note that $1 \leq |q(G^{i-1}(\mathbf{x}))| \leq k$ since $q(G^{i-1}(\mathbf{x}))$ is a sum of $k$ numbers in $\{\pm 1\}$ and $k$ is odd. Likewise, $\|q\|_{\text{co}} \leq \sum_{l=1}^d |w^{i,j}_l| \cdot \|p^i_l\|_{\text{co}} \overset{\|p^i_l\|_{\text{co}} \leq 1}{\leq} \sum_{l=1}^d |w^{i,j}_l| = k$). Since $q$ is $(kL)$-Lipschitz and bounded by $k$, Lemma C.4 implies that $f^*_{i,j}$ is $(K, k, 2k+1, 1/(2kL))$-PTF of $G^{i-1}$ Hence, Theorem C.8 follows from the following lemma and a union bound on the $rq$ different $f^*_{i,j}$.

**Lemma C.9.** *Let $f : \mathcal{X} \to \{\pm 1\}$ be a $(K, M, B, \xi)$-PTF and let $\mathbf{w}^1, \dots, \mathbf{w}^q \in \mathcal{W}_{d,k}$ be independent and uniform. Define $\psi_i(\mathbf{x}) = \mathrm{sign}(\langle \mathbf{w}^i, \mathbf{x} \rangle)$. Then, w.p. $1 - 4d|\mathcal{X}|e^{-\Omega\left(\frac{\xi^2 q}{d}\right)}$ $f$ is $\left(K, O\left(Md^K\right), B\right)$-PTF of $\Psi = (\psi_1, \dots, \psi_q)$.*

*Proof.* Let $W = [\mathbf{w}_1 \cdots \mathbf{w}_q] \in M_{d,q}$ We first show that w.h.p. $W$ approximately reconstruct $\mathbf{x}$ from $\Psi(\mathbf{x})$

*Claim* 2. Let $\alpha_{d,k} = \frac{k}{d} \cdot \frac{\binom{k-1}{(k-1)/2}}{2^{k-1}}$. For any $\mathbf{x} \in \{\pm 1\}^d$ and $\frac{1}{4} \geq \epsilon > 0$ we have $\Pr\left(\left\|\frac{1}{q\alpha_{d,k}} W\Psi(\mathbf{x}) - \mathbf{x}\right\|_\infty \geq \epsilon\right) \leq 4de^{-\Omega\left(\frac{\epsilon^2 q}{d}\right)}$

Before proving the claim, we show that it implies the lemma. Indeed, it implies that w.p. $1 - 4d|\mathcal{X}|e^{-\Omega\left(\frac{\xi^2 q}{d}\right)}$ we have that $\left\|\frac{1}{q\alpha_{d,k}} W\Psi(\mathbf{x}) - \mathbf{x}\right\|_\infty \leq \frac{\xi}{2}$ for any $\mathbf{x} \in \mathcal{X}$. Given this event, we have that

$$1 - \xi \leq \frac{1 - \xi/2}{q\alpha_{d,k}} \left(W\Psi(\mathbf{x}) \odot \mathbf{x}\right)_j \leq 1$$

for any $\mathbf{x} \in \mathcal{X}$ and $j \in [d]$. Thus, if $p : \mathcal{X} \to \mathbb{R}$ is a polynomial hat witness that $f$ is $(K, M, B, \xi)$-PTF, then we have

$$B \geq p\left(\frac{1 - \xi/2}{q\alpha_{d,k}} W\Psi(\mathbf{x})\right) \cdot f(\mathbf{x}) \geq 1$$

Hence, for $q(\mathbf{y}) := p\left(\frac{1-\xi/2}{q\alpha_{d,k}} W\mathbf{y}\right)$ we have that $f$ is $(K, \|q\|_{\mathrm{co}}, B)$-PTF of $\Psi$. By Lemma B.6 and the fact that the norm of each row of $\frac{1-\xi/2}{q\alpha_{d,k}} W$ is at most $\frac{1}{\sqrt{q\alpha_{d,k}}}$ (since the entries of $W$ are in $\{-1, 1, 0\}$) we have

$$\|q\|_{\mathrm{co}} \leq \|p\|_{\mathrm{co}} \cdot \left(\frac{\sqrt{q+1}}{\sqrt{q\alpha_{d,k}}}\right)^K$$

This implies the lemma as $\alpha_{d,k} = \Theta\left(\frac{\sqrt{k}}{d}\right)$ by Lemma B.4.

*Proof.* (of Claim 2) Fix a coordinate $j \in [d]$. It is enough to show that $\Pr\left(\left|\frac{1}{q\alpha_{d,k}} \left(W\Psi(\mathbf{x})\right)_j - x_j\right| \geq \epsilon\right) \leq 4e^{-\Omega\left(\frac{\epsilon^2 q}{d}\right)}$. We note that

$$\frac{1}{q\alpha_{d,k}} \left(W\Psi(\mathbf{x})\right)_j = \frac{1}{q} \sum_{i=1}^q \frac{w_j^i \mathrm{sign}(\langle \mathbf{w}^i, \mathbf{x} \rangle)}{\alpha_{d,k}}$$

Denote $X_i = w_j^i \mathrm{sign}(\langle \mathbf{w}^i, \mathbf{x} \rangle)$. Note that $X_1, \dots, X_q$ are i.i.d. We have

$$
\begin{aligned}
\Pr(X_i = x_j) &= \frac{k}{2d} \left[\Pr(\mathrm{sign}(\langle \mathbf{w}^i, \mathbf{x} \rangle) = 1 | w_j = x_j) + \Pr(\mathrm{sign}(\langle \mathbf{w}^i, \mathbf{x} \rangle) = -1 | w_j = -x_j)\right] \\
&= \frac{k}{2d2^{k-1}} \left[\binom{k-1}{\geq (k-1)/2} + \binom{k-1}{\geq (k-1)/2}\right] \\
&= \frac{k}{2d} \left[1 + \frac{\binom{k-1}{(k-1)/2}}{2^{k-1}}\right]
\end{aligned}
$$

Similarly,

$$
\begin{aligned}
\Pr(X_i = -x_j) &= \frac{k}{2d} \left[\Pr(\mathrm{sign}(\langle \mathbf{w}^i, \mathbf{x} \rangle) - 1 | w_j = x_j) + \Pr(\mathrm{sign}(\langle \mathbf{w}^i, \mathbf{x} \rangle) = 1 | w_j = -x_j)\right] \\
&= \frac{k}{2d2^{k-1}} \left[\binom{k-1}{> (k-1)/2} + \binom{k-1}{> (k-1)/2}\right] \\
&= \frac{k}{2d} \left[1 - \frac{\binom{k-1}{(k-1)/2}}{2^{k-1}}\right]
\end{aligned}
$$

As a result

$$\mathbb{E}X_i = (\Pr(X_i = x_j) - \Pr(X_i = -x_j)) \, x_j = \alpha_{d,k} \cdot x_j$$

And,

$$\Pr(X_i \neq 0) = \Pr(X_i = x_j) + \Pr(X_i = -x_j) = \frac{k}{d}$$

this implies that

$$\frac{\min\left(\Pr(X_i = 1), \Pr(X_i = -1)\right)}{|\mathbb{E}X_i|} = \frac{k}{2d\alpha_{d,k}}\left[1 - \frac{\binom{k-1}{(k-1)/2}}{2^{k-1}}\right] \geq \frac{k}{\alpha_{d,k}4d} \geq \frac{1}{2}$$

and that

$$\frac{|\mathbb{E}X_i|^2}{\Pr(\operatorname{sign}(\langle \mathbf{w}, \mathbf{x} \rangle) w_i \neq 0)} = \frac{k}{d}\left(\frac{\binom{k-1}{(k-1)/2}}{2^{k-1}}\right)^2 \overset{\text{Lemma B.4}}{=} \Theta\left(\frac{1}{d}\right)$$

By Lemma B.3 we have

$$\Pr\left(\left|\frac{1}{q\alpha_{d,k}}\left(W\Psi(\mathbf{x})\right)_j - x_j\right| \geq \epsilon\right) \leq 4e^{-\Omega\left(\frac{\epsilon^2 q}{d}\right)}$$

$\square$

$\square$

## D. Kernels From Random Neurons and Proof of Lemma A.3

Fix a bounded activation $\sigma : \mathbb{R} \to \mathbb{R}$. Given $0 \leq \beta \leq 1$, called the bias magnitude we define a kernel on $\mathbb{R}^n$ by

$$k_{\sigma,\beta,n}(\mathbf{x}, \mathbf{y}) = \mathbb{E}[\sigma(\mathbf{w}^\top \mathbf{x} + b)\sigma(\mathbf{w}^\top \mathbf{y} + b)], \quad b \sim \mathcal{N}(0, \beta^2), \ \mathbf{w} \sim \mathcal{N}\left(0, \frac{1-\beta^2}{n}I_n\right) \tag{22}$$

Note that $\psi((\mathbf{w}, b), \mathbf{x}) = \sigma(\mathbf{w}^\top \mathbf{x} + b)$ is a RFS for $k_{\sigma,\beta,n}$. We next analyze the functions in the corresponding kernel space $\mathcal{H}_{\sigma,\beta,n}$. To this end, we will use the Hermite expansion of $\sigma$ in order to find an explicit expression of $k_{\sigma,\beta,n}$, as well as an explicit embedding $\Psi_{\sigma,\beta,n} : \mathbb{R}^n \to \bigoplus_{s=0}^\infty \left(\mathbb{R}^{n+1}\right)^{\otimes s}$ whose kernel is $k_{\sigma,\beta,n}$. Let

$$\sigma = \sum_{s=0}^\infty a_s h_s \tag{23}$$

be the Hermite expansion of $\sigma$. For $r \geq 1$ denote

$$a_s(r) = \sum_{j=0}^\infty a_{s+2j}\sqrt{\frac{(s+2j)!}{s!}}\frac{(r^2-1)^j}{j!2^j} \tag{24}$$

Note that $a_s(1) = a_s$

**Lemma D.1.** *We have*

$$k_{\sigma,\beta,n}(\mathbf{x}, \mathbf{y}) = \sum_{s=0}^\infty a_s\left(\sqrt{\frac{1-\beta^2}{n}\|\mathbf{x}\|^2 + \beta^2}\right) a_s\left(\sqrt{\frac{1-\beta^2}{n}\|\mathbf{y}\|^2 + \beta^2}\right)\left(\frac{1-\beta^2}{n}\langle\mathbf{x}, \mathbf{y}\rangle + \beta^2\right)^s$$

*Likewise, $k_{\sigma,\beta,n}$ is the kernel of the embedding $\Psi_{\sigma,\beta,n} : \mathbb{R}^n \to \bigoplus_{s=0}^\infty \left(\mathbb{R}^{n+1}\right)^{\otimes s}$ given by*

$$\Psi_{\sigma,\beta,n}(\mathbf{x}) = \left(a_s\left(\sqrt{\frac{1-\beta^2}{n}\|\mathbf{x}\|^2 + \beta^2}\right) \cdot \left[\frac{\sqrt{\frac{1-\beta^2}{n}}\mathbf{x}}{\beta}\right]^{\otimes s}\right)_{s=0}^\infty$$

To prove Lemma D.1 We will use the following Lemma.

**Lemma D.2.** *We have $h_s(ax) = \sum_{j=0}^{\lfloor s/2 \rfloor} \sqrt{\frac{s!}{(s-2j)!}} \frac{a^{s-2j}(a^2-1)^j}{j!2^j} h_{s-2j}(x)$*

*Proof.* By formula (4) we have

$$
\begin{aligned}
\sum_{s=0}^{\infty} \frac{h_s(ax)t^s}{\sqrt{s!}} &= e^{xat - \frac{t^2}{2}} \\
&= e^{xat - \frac{(at)^2}{2} + \frac{(at)^2}{2} - \frac{t^2}{2}} \\
&\overset{\text{Eq. (4)}}{=} e^{\frac{(at)^2}{2} - \frac{t^2}{2}} \left( \sum_{s=0}^{\infty} \frac{h_s(x)a^s t^s}{\sqrt{s!}} \right) \\
&= e^{(a^2-1)\frac{t^2}{2}} \left( \sum_{s=0}^{\infty} \frac{h_s(x)a^s t^s}{\sqrt{s!}} \right) \\
&= \left( \sum_{s=0}^{\infty} \frac{(a^2-1)^s}{s!2^s} t^{2s} \right) \left( \sum_{s=0}^{\infty} \frac{h_s(x)a^s}{\sqrt{s!}} t^s \right) \\
&= \sum_{s=0}^{\infty} \left( \sum_{j=0}^{\lfloor \frac{s}{2} \rfloor} \frac{(a^2-1)^j}{j!2^j} \frac{h_{s-2j}(x)a^{s-2j}}{\sqrt{(s-2j)!}} \right) t^s
\end{aligned}
$$

Thus,

$$
\frac{h_s(ax)}{\sqrt{s!}} = \sum_{j=0}^{\lfloor \frac{s}{2} \rfloor} \frac{(a^2-1)^j}{j!2^j} \frac{a^{s-2j}}{\sqrt{(s-2j)!}} h_{s-2j}(x)
$$

$\square$

*Proof.* (of Lemma D.1) We will prove the formula for $k_{\sigma,\beta,n}$. It is not hard to verify that it implies that $k_{\sigma,\beta,n}$ is the kernel of $\Psi_{\sigma,\beta,n}$ using the fact that $\langle \mathbf{x}^{\otimes s}, \mathbf{y}^{\otimes s} \rangle = \langle \mathbf{x}, \mathbf{y} \rangle^s$. By definition $k_{\sigma,\beta,n}(\mathbf{x}, \mathbf{y}) = \mathbb{E}[\sigma(\mathbf{w}^\top \mathbf{x} + b)\sigma(\mathbf{w}^\top \mathbf{y} + b)]$ where $b \sim \mathcal{N}(0, \beta^2)$ and $\mathbf{w} \sim \mathcal{N}\left(0, \frac{1-\beta^2}{n} I_n\right)$. Let $X = \mathbf{w}^\top \mathbf{x} + b$ and $Y = \mathbf{w}^\top \mathbf{y} + b$. We note that $(X, Y)$ is a centered Gaussian vector with correlation matrix $\begin{pmatrix} \frac{1-\beta^2}{n} \|\mathbf{x}\|^2 + \beta^2 & \frac{1-\beta^2}{n} \langle \mathbf{x}, \mathbf{y} \rangle + \beta^2 \\ \frac{1-\beta^2}{n} \langle \mathbf{x}, \mathbf{y} \rangle + \beta^2 & \frac{1-\beta^2}{n} \|\mathbf{y}\|^2 + \beta^2 \end{pmatrix}$. Denote $r_{\mathbf{x}} = \sqrt{\frac{1-\beta^2}{n} \|\mathbf{x}\|^2 + \beta^2}$ and $r_{\mathbf{y}} = \sqrt{\frac{1-\beta^2}{n} \|\mathbf{y}\|^2 + \beta^2}$. Likewise let $\tilde{X} = \frac{1}{r_{\mathbf{x}}} X$ and $\tilde{Y} = \frac{1}{r_{\mathbf{y}}} Y$. Note that $(X, Y)$ is a centered Gaussian vector with correlation matrix $\begin{pmatrix} 1 & \rho \\ \rho & 1 \end{pmatrix}$ for $\rho = \frac{\frac{1-\beta^2}{n} \langle \mathbf{x}, \mathbf{y} \rangle + \beta^2}{r_{\mathbf{x}} r_{\mathbf{y}}}$ Now, by Lemma D.2 we have

$$
\begin{aligned}
\sigma(rx) &= \sum_{s=0}^{\infty} h_s(rx) \\
&= \sum_{s=0}^{\infty} \left( \sum_{j=0}^{\infty} a_{s+2j} \sqrt{\frac{(s+2j)!}{s!}} \frac{(r^2-1)^j}{j!2^j} \right) r^s h_s(x) \\
&= : \sum_{s=0}^{\infty} a_s(r) r^s h_s(x)
\end{aligned}
$$

Hence,

$$
\begin{aligned}
k_{\sigma,\beta,n}(\mathbf{x},\mathbf{y}) \quad &= \quad \mathbb{E}\sigma(r_{\mathbf{x}}\tilde{X})\sigma(r_{\mathbf{y}}\tilde{Y}) \\
&= \quad \sum_{i=0}^{\infty}\sum_{j=0}^{\infty} a_i(r_{\mathbf{x}})r_{\mathbf{x}}^i a_j(r_{\mathbf{y}})r_{\mathbf{x}}^j \mathbb{E}h_i(\tilde{X})h_j(\tilde{Y}) \\
&\overset{\text{Eq. (6)}}{=} \quad \sum_{s=0}^{\infty} a_s(r_{\mathbf{x}})r_{\mathbf{x}}^s a_s(r_{\mathbf{y}})r_{\mathbf{y}}^s \rho^s \\
&= \quad \sum_{s=0}^{\infty} a_s(r_{\mathbf{x}})a_s(r_{\mathbf{y}}) \left( \frac{1-\beta^2}{n}\langle\mathbf{x},\mathbf{y}\rangle + \beta^2 \right)^s
\end{aligned}
$$

$\square$

**Lemma D.3.** *Let $r > 0$ such that $|1 - r^2| =: \epsilon < \frac{1}{2}$. We have*

$$
|a_s(r) - a_s(1)| \le \|\sigma\|2^{(s+2)/2}\frac{\epsilon}{\sqrt{1-2\epsilon^2}}
$$

*Proof.* We have

$$
\begin{aligned}
|a_s(r) - a_s(1)| \qquad &= \qquad \left| \sum_{j=1}^{\infty} a_{s+2j}\sqrt{\frac{(s+2j)!}{s!}}\frac{(r^2-1)^j}{j!2^j} \right| \\
&\overset{\text{Cauchy-Schwartz and }\|\sigma\|=\sqrt{\sum_{i=0}^{\infty}a_i^2}}{\le} \quad \|\sigma\|\sqrt{\sum_{j=1}^{\infty}\frac{(s+2j)!}{s!}\frac{(r^2-1)^{2j}}{(j!)^2 2^{2j}}} \\
&\overset{(2j)!\le(j!2^j)^2}{\le} \quad \|\sigma\|\sqrt{\sum_{j=1}^{\infty}\frac{(s+2j)!}{s!(2j)!}(r^2-1)^{2j}} \\
&= \quad \|\sigma\|\sqrt{\sum_{j=1}^{\infty}\binom{s+2j}{s}(r^2-1)^{2j}} \\
&\le \quad \|\sigma\|\sqrt{\sum_{j=1}^{\infty}2^{s+2j}(r^2-1)^{2j}} \\
&= \quad \|\sigma\|2^{s/2}\sqrt{\sum_{j=1}^{\infty}(2r^2-2)^{2j}} \\
&= \quad \|\sigma\|2^{s/2}|2r^2-2|\frac{1}{\sqrt{1-(2r^2-2)^2}}
\end{aligned}
$$

$\square$

**Lemma D.4.** *Assume that $1 - \beta^2 < \frac{1}{2}$ for $\beta > 0$. Let $\mathcal{X} \subseteq [-1,1]^n$. Let $p : \mathcal{X} \to \mathbb{R}$ be a degree $K$ polynomial. Let $K' \ge K$. There is $g \in \mathcal{H}_{\sigma,\beta,n}(\mathcal{X})$ such that*

1. $g(\mathbf{x}) = \dfrac{a_{K'}\left(\sqrt{\frac{1-\beta^2}{n}\|\mathbf{x}\|^2 + \beta^2}\right)}{a_{K'}}p(\mathbf{x})$

2. $\|g\|_{\sigma,\beta,n} \le \dfrac{1}{a_{K'}\beta^{K'-K}}\left(\dfrac{n}{1-\beta^2}\right)^{K/2}\|p\|_{\mathrm{co}}$

3. $\|g - p\|_{\infty} \le \|p\|_{\infty}\dfrac{\|\sigma\|}{a_{K'}}2^{(K'+2)/2}\dfrac{1-\beta^2}{\sqrt{1-2(1-\beta^2)^2}}$

*Proof.* Write $p(\mathbf{x}) = \sum_{\alpha \in \{0,\dots,K\}^n, \|\alpha\|_1 \le K} b_\alpha \mathbf{x}^\alpha$. For $\alpha \in \{0,\dots,K\}^n$, $\|\alpha\|_1 \le K$ we let $\tilde{\alpha} \in [n+1]^{K'}$ be a sequence such that for any $i \in [n]$ we have $\tilde{\alpha}_j = i$ for exactly $\alpha_i$ indices $j \in [K']$ and $\tilde{\alpha}_j = n+1$ for the remaining $K' - \|\alpha\|_1$ indices. Let $A \in (\mathbb{R}^{n+1})^{\otimes K'} \subseteq \bigoplus_{s=0}^\infty (\mathbb{R}^{n+1})^{\otimes s}$ be the tensor

$$
A_\gamma = \begin{cases} \frac{1}{a_{K'}\beta^{K'-\|\alpha\|_1}} \left(\frac{n}{1-\beta^2}\right)^{\|\alpha\|_1/2} b_\alpha & \gamma = \tilde{\alpha} \text{ for some } \alpha \\ 0 & \text{otherwise} \end{cases}
$$

and let

$$
g(\mathbf{x}) = \langle A, \Psi_{\sigma,\beta,n}(\mathbf{x}) \rangle
$$

It is not hard to verify that $g(\mathbf{x}) = \frac{a_{K'}\left(\sqrt{\frac{1-\beta^2}{n}\|\mathbf{x}\|^2 + \beta^2}\right)}{a_{K'}} p(\mathbf{x})$. By Theorem B.9 $g \in \mathcal{H}_{\sigma,\beta,n}$ and satisfies $\|g\|_{\sigma,\beta,n} \le \|A\|$.

Finally, since $\frac{1}{\beta^{K'-\|\alpha\|_1}}\left(\frac{n}{1-\beta^2}\right)^{\|\alpha\|_1/2} \le \frac{1}{\beta^{K'-K}}\left(\frac{n}{1-\beta^2}\right)^{K/2}$ we have $\|A\| \le \frac{1}{a_{K'}\beta^{K'-K}}\left(\frac{n}{1-\beta^2}\right)^{K/2}\|p\|_{\text{co}}$. We therefore proved the first and the second items. To prove the last item we note that for any $\mathbf{x} \in \mathcal{X}$ we have

$$
\begin{aligned}
|g(\mathbf{x}) - p(\mathbf{x})| &= |p(\mathbf{x})| \cdot \left| \frac{a_{K'}\left(\sqrt{\frac{1-\beta^2}{n}\|\mathbf{x}\|^2 + \beta^2}\right)}{a_{K'}} - 1 \right| \\
&= \frac{|p(\mathbf{x})|}{a_{K'}} \left| a_{K'}\left(\sqrt{\frac{1-\beta^2}{n}\|\mathbf{x}\|^2 + \beta^2}\right) - a_{K'} \right|
\end{aligned}
$$

Define $r = \sqrt{\frac{\|\mathbf{x}\|^2}{n}(1-\beta^2) + \beta^2}$ and note that since $0 \le \|\mathbf{x}\|^2 \le n$ we have

$$
\beta^2 \le r^2 \le 1 \Rightarrow \epsilon := |1 - r^2| \le 1 - \beta^2 < \frac{1}{2}
$$

Hence, by Lemma D.3 we have

$$
|g(\mathbf{x}) - p(\mathbf{x})| \le \frac{|p(\mathbf{x})|}{a_{K'}}\|\sigma\|2^{(K'+2)/2}\frac{1-\beta^2}{\sqrt{1-2(1-\beta^2)^2}}
$$

which proves the last item $\qquad\square$

Combining with Lemma D.4 with Lemma B.11 we get

**Lemma D.5.** *Assume that* $1 - \beta^2 < \frac{1}{2}$ *for* $\beta > 0$. *Let* $\mathcal{X} \subset [-1,1]^n$. *Fix a degree $K$ polynomial* $p : \mathcal{X} \to [-1,1]$ *and* $K' \ge K$. *Let* $(W, \mathbf{b}) \in \mathbb{R}^{q \times n} \times \mathbb{R}^q$ *be* $\beta$-*Xavier pair. Then there is a vector* $\mathbf{w} = \mathbf{w}(W, \mathbf{b}) \in \mathbb{R}^q$ *such that*

$$
\forall \mathbf{x} \in \mathcal{X}, \ \Pr\left(|\langle \mathbf{w}, \sigma(W\mathbf{x}+\mathbf{b})\rangle - p(\mathbf{x})| \ge \epsilon + \frac{\|\sigma\|}{a_{K'}}2^{(K'+2)/2}\frac{1-\beta^2}{\sqrt{1-2(1-\beta^2)^2}}\right) \le \delta
$$

*for*

$$
\delta = 2\exp\left(-q \cdot \frac{a_{K'}^4\beta^{4K'-4K}(1-\beta^2)^{2K}\epsilon^4}{32n^{2K}\|p\|_{\text{co}}^4\|\sigma\|_\infty^4}\right)
$$

*Moreover*

$$
\|\mathbf{w}\| \le \frac{2\|\sigma\|_\infty}{\epsilon\sqrt{q}} \cdot \frac{1}{a_{K'}^2\beta^{2K'-2K}}\left(\frac{n}{1-\beta^2}\right)^K\|p\|_{\text{co}}^2
$$

We next specialize Lemma D.5 for the needs of our paper and explain how it implies Lemma A.3. Recall that for $\epsilon > 0$ we defined $\frac{3}{4} \le \beta_{\sigma,K',K}(\epsilon) < 1$ as the minimal number such that if $\beta_{\sigma,K',K}(\epsilon) \le \beta < 1$ then

$$
\frac{\|\sigma\|}{a_{K'}}2^{(K'+2)/2}\frac{1-\beta^2}{\sqrt{1-2(1-\beta^2)^2}} \le \frac{\epsilon}{2}
$$

We also defined

$$\delta_{\sigma,K',K}(\epsilon, \beta, q, M, n) = \begin{cases} 1 & \frac{4\|\sigma\|_\infty}{\epsilon\sqrt{q}} \cdot \frac{1}{a_{K'}^2 \beta^{2K'-2K}} \left(\frac{n}{1-\beta^2}\right)^K M^2 > 1 \\ 2\exp\left(-q \cdot \frac{a_{K'}^4 \beta^{4K'-4K}(1-\beta^2)^{2K}\epsilon^4}{512 n^{2K} M^4 \|\sigma\|_\infty^4}\right) & \text{otherwise} \end{cases}$$

We can now prove Lemma A.3 restated which we restate next.

**Lemma D.6.** *(Lemma A.3 restated) Fix $\mathcal{X} \subset [-1,1]^n$, a degree $K$ polynomial $p : \mathcal{X} \to [-1,1]$, $K' \geq K$ and $\epsilon > 0$. Let $(W, \mathbf{b}) \in \mathbb{R}^{q \times n} \times \mathbb{R}^q$ be $\beta$-Xavier pair for $1 > \beta \geq \beta_{\sigma,K',K}(\epsilon)$. Then there is a vector $\mathbf{w} = \mathbf{w}(W, \mathbf{b}) \in \mathbb{B}^q$ such that*

$$\forall \mathbf{x} \in \mathcal{X}, \ \Pr\left(|\langle \mathbf{w}, \sigma(W\mathbf{x} + \mathbf{b})\rangle - p(\mathbf{x})| \geq \epsilon\right) \leq \delta_{\sigma,K',K}(\epsilon, \beta, q, \|p\|_{\mathrm{co}}, n)$$

*Proof.* Fix $\mathbf{x} \in \mathcal{X}$. By Lemma D.5 there is a vector $\mathbf{v} \in \mathbb{R}^q$ such that

$$\Pr\left(|\langle \mathbf{v}, \sigma(W\mathbf{x}+\mathbf{b})\rangle - p(\mathbf{x})| \geq \epsilon\right) \leq \Pr\left(|\langle \mathbf{v}, \sigma(W\mathbf{x}+\mathbf{b})\rangle - p(\mathbf{x})| \geq \frac{\epsilon}{2} + \frac{\|\sigma\|}{a_{K'}} 2^{(K'+2)/2} \frac{1-\beta^2}{\sqrt{1-2(1-\beta^2)^2}}\right) \leq \delta \quad (25)$$

for

$$\delta = 2\exp\left(-q \cdot \frac{a_{K'}^4 \beta^{4K'-4K}(1-\beta^2)^{2K}\epsilon^4}{512 n^{2K} \|p\|_{\mathrm{co}}^4 \|\sigma\|_\infty^4}\right)$$

Moreover

$$\|\mathbf{v}\| \leq \frac{4\|\sigma\|_\infty}{\epsilon\sqrt{q}} \cdot \frac{1}{a_{K'}^2 \beta^{2K'-2K}} \left(\frac{n}{1-\beta^2}\right)^K \|p\|_{\mathrm{co}}^2$$

Define $\mathbf{w}$ to be the projection of $\mathbf{v}$ on $\mathbb{B}^d$. We now split into cases. If $\frac{4\|\sigma\|_\infty}{\epsilon\sqrt{q}} \cdot \frac{1}{a_{K'}^2 \beta^{2K'-2K}} \left(\frac{n}{1-\beta^2}\right)^K \|p\|_{\mathrm{co}}^2 \leq 1$ then $\mathbf{v} = \mathbf{w}$ and $\delta = \delta_{\sigma,K',K}(\epsilon, \beta, q, \|p\|_{\mathrm{co}}, n)$, so the Lemma follows from Equation (25). Otherwise, we have $\delta_{\sigma,K',K}(\epsilon, \beta, q, \|p\|_{\mathrm{co}}, n) = 1$ and the Lemma is trivially true. $\square$

