# OpenReview forum: "Deep Networks Learn Deep Hierarchical Models"
_ICML.cc/2026/Conference — ICML 2026 regular_

### Official Review · Reviewer_AHsp · 2026-03-06

**Soundness:** 3
**Presentation:** 3
**Significance:** 3
**Originality:** 3
**Overall Recommendation:** 4
**Confidence:** 1

**Summary:**

This paper analyzes a class of hierarchical models in which labels are structured through a nested sequence. The authors prove that layer-wise stochastic gradient descent (SGD) applied to deep residual networks can efficiently learn such models.

**Compliance With Llm Reviewing Policy:**

Affirmed.

**Final Justification:**

While I believe this paper provides theoretical insights, its central hierarchy assumption seems primarily theoretical and difficult to justify in practice. As the authors themselves acknowledge, an evaluation of layer-wise SGD is not meaningful unless the hierarchy assumption is first established. For these reasons, I keep my original score.

**Key Questions For Authors:**

1. It would strengthen the paper to discuss the most relevant theoretical results on residual networks and layer-wise learning, and to clearly distinguish how the present framework differs from them.

2. Are there real-world tasks that genuinely satisfy the assumed hierarchical model? Providing concrete examples would help clarify the applicability of the theoretical framework.

**Limitations:**

Yes.

**Strengths And Weaknesses:**

**Soundness**: The submission is technically sound. The main claims are supported by rigorous theoretical analysis, and the proofs appear correct under the stated assumptions. The methods are appropriate. The authors list out strong modeling assumptions. The paper would benefit from simple experiments on the effectiveness of the layer-wise SGD.

**Presentation**: The submission is clearly written and well structured. The inclusion of the proof sketch helps guide the reader through the main ideas. The paper would benefit from more explicitly positioning the work relative to prior theoretical results on residual networks and layer-wise training.

**Significance**: The paper addresses an important theoretical question concerning the learnability of hierarchical models and the role of layer-wise training in residual networks. It advances understanding of how hierarchical structure can be exploited by deep architectures and may influence future research in deep learning theory.

**Originality**: The paper offers new theoretical insight by formalizing a class of deep hierarchical models and proving that layer-wise training of residual networks can efficiently learn them. The combination of random feature theory and hierarchical modeling is nontrivial. The novelty is justified, though clearer positioning relative to closely related theoretical work on deep/residual networks and layer-wise training would strengthen the contribution.

---

> ### Author Rebuttal · Authors · 2026-03-28
>
> Dear Reviewer,
>
> Thank you for your constructive review and for recognizing the theoretical significance and originality of our work!
>
> We will incorporate your suggestions as follows:
>
> 1. Positioning Relative to Prior Theoretical Results. We will expand our related work section to make this distinction clearer. Prior provable guarantees for ResNets primarily cover functions expressible by shallow or logarithmic-depth networks. Our framework uniquely proves that layer-wise training can efficiently learn hierarchical models requiring polynomial depth, pushing theoretical understanding to the limit of efficient learnability.
>
> 2. Real-World Examples & The Hierarchical Assumption. Conceptually, pipelines like computer vision map nicely to our hierarchy ($L_1$: edges from pixels $\to$ $L_2$: shapes $\to$ $L_3$: object parts). However, we will emphasize in the revision that at the moment, this is a purely theoretical work. In order to explicitly test this hierarchical assumption in practice, more progress—such as developing more refined notions of hierarchies—needs to be made.
>
> 3. Empirical Evaluation. Because our contribution is foundational theory, establishing these polynomial-depth mathematical bounds requires substantial machinery and stands as an independent contribution. As noted above, before we can provide meaningful empirical evaluations of layer-wise SGD on this specific framework, further refinements of the formal hierarchy are needed. We will explicitly highlight bridging this gap to empirical testing as a key direction for future work.
>
> Thank you again for your insightful suggestions!

---

> > ### Author Rebuttal · Reviewer_AHsp · 2026-04-02
> >
> > While I believe this paper provides theoretical insights, its central hierarchy assumption seems primarily theoretical and difficult to justify in practice. As the authors themselves acknowledge, an evaluation of layer-wise SGD is not meaningful unless the hierarchy assumption is first established. For these reasons, I keep my original score.

---

> > > ### Author Response · Authors · 2026-04-02
> > >
> > > Thank you for your continued engagement with our work and for sharing your perspective.
> > >
> > > We respectfully ask that you evaluate our submission as a theoretical paper, which is what it is. As is widely recognized in the field, deep learning theory is currently significantly behind the empirical practice of deep learning. Because of this gap, virtually all published theoretical papers that establish provable learnability by SGD on neural networks rely on idealized assumptions that are not immediately tested—or even easily testable—in empirical settings.
> > >
> > > To contextualize our contribution within this theoretical landscape: the vast majority of existing SOTA works rely on highly restrictive assumptions that only prove learnability for shallow, depth-two networks. In contrast, our hierarchical assumption is significantly more relaxed. It allows us to mathematically demonstrate, for the first time, how layer-wise SGD can efficiently learn deep hierarchical models requiring polynomial depth.
> > >
> > > By breaking past the shallow and log-depth barriers, our work significantly advances the theoretical state-of-the-art. While we agree with you that bridging these theoretical models to empirical practice is a crucial next step for the field, we hope you might consider judging the paper based on its fundamental mathematical advancement and the new theoretical understanding it brings to deep networks.
> > >
> > > Thank you again for your time, effort, and valuable feedback!

---

### Official Review · Reviewer_6Hav · 2026-03-06

**Soundness:** 3
**Presentation:** 3
**Significance:** 3
**Originality:** 3
**Overall Recommendation:** 4
**Confidence:** 1

**Summary:**

This paper studies the ability of deep networks, especially residual networks trained with layerwise SGD, to efficiently learn a family of hierarchical supervised multi-label models. It assumes an unknown label hierarchy where lower-level labels are simple functions of the input, and higher-level labels depend on simpler lower ones. The core claim is that layerwise SGD on ResNets can efficiently learn arbitrarily deep hierarchical structures, exceeding prior deep learning theory bounded by log-depth circuits and achieving the depth limit of efficient learnability.

**Compliance With Llm Reviewing Policy:**

Affirmed.

**Final Justification:**

No further comments.

**Key Questions For Authors:**

See weaknesses.

**Limitations:**

Yes.

**Strengths And Weaknesses:**

**Strengths**:

- This paper is comprehensive with formal definitions and detailed proofs.
- The discussion connects this work to both learning theory and broader scientific discourse, inspiring novel directions for future investigation.
- Although this paper is mathematically dense, it is well structured with explicit limitations.

**Weaknesses**:

- No experiments or empirical evaluation is provided in this paper. If the authors can provide experimental results, the claims in this paper would be more credible.
- The loss function adopted in this paper is non-standard, which differs significantly from the loss commonly used in practice.
- The assumptions underlying the layerwise training scheme are overly restrictive, leading to a substantial gap between the theoretical setup and the joint optimization employed in real-world practice.
- The definition of \circ is missing.

---

> ### Author Rebuttal · Authors · 2026-03-27
>
> Dear Reviewer,
>
> Thank you for recognizing our paper as comprehensive, well-structured, and inspiring!
>
> We address your points below:
>
> 1. Lack of Experiments: Our contribution is strictly theoretical. Establishing the first rigorous mathematical proof that neural networks can efficiently learn polynomial-depth hierarchies (surpassing prior log-depth SOTA) requires heavy mathematical machinery. We believe this foundational guarantee stands strongly as an independent contribution.
>
> 2. Non-standard Loss & Layerwise Training: We agree these differ from empirical practice, but they are necessary theoretical abstractions. Analyzing standard losses and joint optimization for arbitrarily deep networks currently presents intractable mathematical roadblocks. Specifically, layerwise training allows us to rigorously guarantee that previously learned lower-level representations are not catastrophically forgotten. We will explicitly clarify this gap in the text and highlight joint optimization as a critical direction for future work.
>
> 3. Missing Definition: We will define $\circ$ as function composition. Thank you for catching this!
>
> Thank you again for your constructive feedback and support!

---

> > ### Author Rebuttal · Reviewer_6Hav · 2026-04-03
> >
> > Thank you to the authors for their rebuttal. I appreciate the effort put into addressing my questions and concerns. While this work does make contributions from a theoretical perspective, the lack of empirical evidence remains a major concern. Therefore, I will retain my score.

---

> > > ### Author Response · Authors · 2026-04-03
> > >
> > > Dear Reviewer 6Hav,
> > >
> > > Thank you for your continued engagement with our work.
> > >
> > > We respectfully ask that you consider evaluating our submission primarily as a theoretical paper. In the subfield of deep learning theory, proving learnability by SGD is notoriously difficult. As a result, virtually all published theoretical works rely on extremely strong assumptions—most commonly, assuming that the data is realizable by shallow, depth-two networks. Not only are these standard assumptions rarely tested empirically in such papers, but it is widely acknowledged that they clearly do not hold in practical deep learning.
> > >
> > > Our paper takes a significant step forward by relaxing these restrictive assumptions, allowing us to mathematically prove learnability for deep, polynomial-depth hierarchies for the first time.
> > >
> > > Because deep learning theory is currently far behind practice, mathematical models are necessarily idealized. We believe that advancing the theoretical boundary of what deep networks can provably learn is a substantial independent contribution, even if empirical testing of these exact theoretical models remains an open challenge for the broader field.
> > >
> > > We hope you might factor the context of this theoretical landscape into your final assessment. Thank you again for your time, effort, and valuable feedback!

---

### Official Review · Reviewer_hKUs · 2026-03-11

**Soundness:** 3
**Presentation:** 2
**Significance:** 2
**Originality:** 4
**Overall Recommendation:** 3
**Confidence:** 2

**Summary:**

This paper studies supervised multi-label deep learning problems where the labels have a hierarchical structure such that deeper labels are simple functions (specifically PTFs) of prior labels and the base level labels are simple functions of the model input. They consider a residual network that is trained layer by layer using SGD and prove that this model can learn an arbitrary function of the aforementioned hierarchical form using a number of samples and a model size that is polynomially bounded.

**Compliance With Llm Reviewing Policy:**

Affirmed.

**Final Justification:**

The speculative foundation of the problem makes it difficult for me to see the relevance of this work. Further, I felt that the presentation could be improved significantly to avoid confusing the reader and to focus more heavily on the theoretical contributions, rather than the weak connections to real-world deep-learning data. I would urge the authors to keep their discussion of real-world applications more focused and avoid overstating the significance of this work.

With that said, as discussed in the rebuttal, the theoretical insights into this problem setting are interesting and well defended. As such, I've increased my overall score to a Weak Reject.

**Key Questions For Authors:**

1. Could you clarify the "brain dump" hierarchy you discuss in 3.1? Are you just saying that any question asked of a human brain is a simple function away from the answer? How would you expect the "brain dump" direction to be used in future theoretical or practical work?
2. In your abstract, you say that "the mere existence of human teachers [support] the hypothesis that hierarchical structures are inherently available." What exactly do you mean by this? Is this just referencing the brain dump, as the labels are provided by a human, or is there a different "human teacher?" Could you better clarify how their "existence supports the hypothesis"?
3. In your conclusion, what do you mean by "a middle ground between Software Engineering and Learning?" I don't understand how your class of function or problem setting have anything to do with software engineering.

**Limitations:**

Yes

**Strengths And Weaknesses:**

__Soundness__

This main theorem of this paper seems well structured and the assumptions, such as needing a specific layerwise training and a specific hierarchical class of function, are explicitly identified. Much of this work requires an understanding of learning theory that is outside my expertise so I cannot speak to the soundness of the proofs, but the results appear to be technically plausible.

Beyond this proof, however, I found much of the work to be very speculative and confusing. A significant portion of the introduction is used for giving perspectives on various deep learning problems throughout history. You do not demonstrate any connection between these problems and the hierarchical labels you consider beyond of saying they are "very complex" and therefore "likely to possess a hierarchical structure." Similarly, I was quite confused about the "brain dump" hierarchy in 3.1 and what relevance it has to the rest of the paper. It seems like you are trying to justify your assumptions by making claims about how hierarchical data may be manifested in real world data, but it seems to be entirely speculation. I think these ideas would make a lot more sense in a discussion section or with some empirical evidence that real data can meaningfully match your assumptions. Furthermore, I don't understand what point you are making about "human teachers." Is this meant to reference the "brain-dump" ideas, or something entirely different?

__Presentation__

The presentation of the paper could be improved to help readability. In the introduction, your contributions could be more concisely laid out, as they currently are spread across long meandering paragraphs. A dedicated discussion sections would make the speculative parts of your paper easier to put in context and less distracting from your main mathematical results. The conclusion and future work section is also somewhat confusing. The perspectives you lay out, particularly in the "Software Engineering" and "A modified narrative for learning theory," are not well argued, and it's difficult to see what point you are trying to make. Meanwhile, the limitations feel awkwardly attached at the end when they should likely have their own dedicated section. Finally, there are a few typos (section 4 before equation 7, the citation in 5.2, etc).

__Significance__

The theoretical contributions to learning theory seem interesting, though I am not familiar enough with this domain to know how this work impacts the greater field. With that said, the connections to common real-world deep learning problems are unconvincing and it remains unclear to me how one might apply these findings in practice.

__Originality__

The paper provides a novel theoretical framework that differs from prior work by considering hierarchical labels and using models that require polynomial depth. The connections between hierarchical label settings and other deep-learning problems are also quite novel.

---

> ### Author Rebuttal · Authors · 2026-03-27
>
> Dear Reviewer,
>
> Thank you for reviewing our paper. We acknowledge that our broader framework is admittedly a bit speculative and theoretical, with no immediate practical implications. However, we believe ICML is exactly the right venue to accommodate and foster such forward-looking foundational work. Furthermore, we completely agree that our core mathematical results should not be overshadowed by this speculative motivation, and we will adopt your structural suggestions.
>
> 1. and 2. "Brain Dump" & "Human Teachers". This simply refers to how humans generate real-world data (e.g., text, tutorials) containing rich intermediate reasoning steps and sub-labels. These intermediate steps act as the "intermediate supervision" our framework relies on. As you suggested, we will remove this entirely from the introduction and move it to a clearly labeled "Discussion/Motivation" section.
>
> 3. "Software Engineering vs. Learning"This was intended as an analogy. Pure software engineering requires manually coding every intermediate step; pure ML learns everything end-to-end. Our framework is a "middle ground" where humans provide intermediate concepts (the hierarchy of labels), and the network learns the simple functions between them. We will clarify this and relocate it to the discussion section.
>
>
> Thank you for helping us improve the paper's clarity!

---

> > ### Author Rebuttal · Reviewer_hKUs · 2026-04-02
> >
> > I find the connection to real data to be much too speculative and underbaked to be considered "forward-looking." You talk abstractly about complex data across many unspecified domains and assume that they can all be represented by a very simple hierarchical circuit. At the very least, it would be beneficial to consider a specific type of problem or domain to ground your theory. As it stands, there is no way to verify if these assumptions hold.
> >
> > Without this connection, it is unclear why we should care about this class of problem, making the significance of your theoretical contributions quite limited.
> >
> > Finally, the connection to "software engineering" remains confusing and poorly argued. It seems that the only connection is that one can learn the simplified circuit of the "brain" easier than the "brain" itself provided they have access to the intermediate steps. This connection doesn't make much sense and I don't see what value it adds to the paper.
> >
> > To address these issues would require an extensive reworking of the paper. As such, I've elected to keep my score the same.

---

> > > ### Author Response · Authors · 2026-04-03
> > >
> > > Thank you for your continued engagement with our work and for sharing your perspective.
> > >
> > > We reiterate our response to reviewer AHsp
> > >
> > > We respectfully ask that you evaluate our submission as a theoretical paper, which is what it is. As is widely recognized in the field, deep learning theory is currently significantly behind the empirical practice of deep learning. Because of this gap, virtually all published theoretical papers that establish provable learnability by SGD on neural networks rely on idealized assumptions that are not immediately tested—or even easily testable—in empirical settings.
> > >
> > > To contextualize our contribution within this theoretical landscape: the vast majority of existing SOTA works rely on highly restrictive assumptions that only prove learnability for shallow, depth-two networks. In contrast, our hierarchical assumption is significantly more relaxed. It allows us to mathematically demonstrate, for the first time, how layer-wise SGD can efficiently learn deep hierarchical models requiring polynomial depth.
> > >
> > > By breaking past the shallow and log-depth barriers, our work significantly advances the theoretical state-of-the-art. While we agree with you that bridging these theoretical models to empirical practice is a crucial next step for the field, we hope you might consider judging the paper based on its fundamental mathematical advancement and the new theoretical understanding it brings to deep networks.
> > >
> > > Thank you again for your time, effort, and valuable feedback!

---

### Official Review · Reviewer_zshr · 2026-03-13

**Soundness:** 3
**Presentation:** 3
**Significance:** 3
**Originality:** 4
**Overall Recommendation:** 6
**Confidence:** 2

**Summary:**

This work addresses a central challenge of machine learning: identifying models that is both rich enough to represent data and provably learnable by standard gradient-based optimization. The authors introduce a class of "hierarchical model" where labels are organized into levels. The innovation of this work is that layer wise gradient descent on residual networks can efficiently learn this hierarchical class even when the underlying hierarchy is unknown. This work positions hierarchical structures as a potential theoretical basis for the empirical success of deep learning

**Compliance With Llm Reviewing Policy:**

Affirmed.

**Key Questions For Authors:**

- It would be very helpful to the reader to include a conceptual diagram showing how the labels at different levels of the hierarchy interact during the layer wise SGD process
- While the results rely on access to supervision for intermediate levels of the hierarchy (i.e., labels for internal nodes of the hierarchy in addition to the final prediction target), in many application settings such intermediate supervision is not available. Could the authors comment on how essential this is for the analysis

**Limitations:**

Yes

**Strengths And Weaknesses:**

- The theoretical contribution is substantial: the authors extend provable learnability results to hierarchical models that may require polynomial depth, advancing beyond prior work that focused primarily on shallow or logarithmic-depth function classes.
- The hierarchical labeling framework studied in this paper is clearly defined and provides a clean abstraction for compositional structure in real-world learning problems, including in vision and biomedical systems.
- The authors provide a concrete discussion of limitations and future directions, including discussing where the assumptions of the framework are strong, such as the strong supervision assumption, and extending the theory to systems where all layers are trained simultaneously. The author notes that, in fact in practice the layers are learned jointly, and I do think further commentary on this would be interesting and helpful to a practitioner.

---

> ### Author Rebuttal · Authors · 2026-03-27
>
> Dear Reviewer,
>
> Thank you for your strong support of our work and for the constructive suggestions! We address your points below and will incorporate all of them into the revised manuscript.
>
> 1. Conceptual Diagram. We completely agree. We will add a diagram in Section 2 that visually traces the layer-wise SGD process. It will illustrate how base inputs learn $L_1$ labels, and how residual connections carry these representations forward so subsequent layers can seamlessly learn higher-level labels ($L_2$, $L_3$, etc.) without forgetting lower-level features.
>
> 2. The Necessity of Intermediate Supervision. You raise an excellent point. While intermediate supervision allows us to bypass computational hardness, dense supervision is actually not strictly necessary. Within our framework, we can drop intermediate levels while maintaining polynomial learnability, as long as we do not drop more than $\omega(1)$ consecutive levels. Doing so increases the training cost but it remains polynomial. We will add a paragraph clarifying this nuance, as it better reflects real-world applications where intermediate labels might be sparse.
>
> 3. Joint vs. Layer-wise Training. We rely on layer-wise training in our proofs to rigorously guarantee that previously learned representations are preserved as the network deepens. To better serve practitioners, we will expand our discussion section to hypothesize how practical joint training might implicitly follow a similar curriculum—where early layers converge to simpler concepts before deeper layers do.
>
> Thank you again for your thorough review and advocacy for our paper!

---

### Decision · Program_Chairs · 2026-04-30

**Decision:**

Accept (regular)

**Comment:**

This theoretical paper proves learnability with SGD of a hierarchy of problems. The theorems seem novel, interesting, and rigorous; I am happy to recommend its acceptance to ICML.

While the setting is different, there is some similarity here to results about the learnability of problems with intermediate "chain of thought" labels in autoregressive models. In particular, see [A Theory of Learning with Autoregressive Chain of Thought](https://arxiv.org/abs/2503.07932). The final version of this paper would benefit from a discussion of the relationship to this work, as well as the various issues previously discussed with the reviewers; please incorporate these.